# DETree: DEtecting Human-AI Collaborative Texts via Tree-Structured Hierarchical Representation Learning

Yongxin He[*1,4]    Shan Zhang[*2,3,4]    Yixuan Cao[†1,4]    Lei Ma[2,3,4]    Ping Luo[1,4]

[1]State Key Laboratory of AI Safety, Institute of Computing Technology, Chinese Academy of Sciences, Beijing, China
[2]The Key Laboratory of Cognition and Decision Intelligence for Complex Systems, Institute of Automation, Chinese Academy of Sciences, Beijing, China
[3]School of Artificial Intelligence, University of Chinese Academy of Sciences, Beijing, China
[4]University of Chinese Academy of Sciences, CAS, Beijing, China

## Abstract

Detecting AI-involved text is essential for combating misinformation, plagiarism, and academic misconduct. However, AI text generation includes diverse collaborative processes (AI-written text edited by humans, human-written text edited by AI, and AI-generated text refined by other AI), where various or even new LLMs could be involved. Texts generated through these varied processes exhibit complex characteristics, presenting significant challenges for detection. Current methods model these processes rather crudely, primarily employing binary classification (purely human vs. AI-involved) or multi-classification (treating human-AI collaboration as a new class). We observe that representations of texts generated through different processes exhibit inherent clustering relationships. Therefore, we propose DETree, a novel approach that models the relationships among different processes as a Hierarchical Affinity Tree structure, and introduces a specialized loss function that aligns text representations with this tree. To facilitate this learning, we developed RealBench, a comprehensive benchmark dataset that automatically incorporates a wide spectrum of hybrid texts produced through various human-AI collaboration processes. Our method improves performance in hybrid text detection tasks and significantly enhances robustness and generalization in out-of-distribution scenarios, particularly in few-shot learning conditions, further demonstrating the promise of training-based approaches in OOD settings. Our code and dataset are available at `https://github.com/heyongxin233/DETree`.

## 1   Introduction

With the widespread deployment of large language models (LLMs) [1, 2, 3, 4, 5], AI-involved text generation has become increasingly diverse, encompassing LLM-assisted refinement of human drafts, human revision of LLM outputs, and collaborative generation involving multiple LLMs, among other strategies. In practice, tolerance for AI involvement varies across scenarios. For example, LLM-based editing may be acceptable in copywriting, whereas it is strictly prohibited in originality-focused contexts. This necessitates detection methods that can identify not only whether AI is involved, but also how it is involved and to what extent. Despite this need, existing AI-text detection methods [6, 7, 8, 9] are typically developed and evaluated on specific datasets, focusing on binary

---

[*]Equal contribution.
[†]Corresponding author: `caoyixuan@ict.ac.cn`

39th Conference on Neural Information Processing Systems (NeurIPS 2025).

classification between purely human-written and purely machine-generated content. However, such approaches often struggle to generalize to complex scenarios involving human–AI collaborative text.

Although recent studies have attempted to detect human–AI collaborative text, such as by regressing the degree of LLM involvement [10] or classifying text based on content and style features from predefined prompts [11]. However, these methods rely on coarse-grained estimation of AI involvement or shallow statistical representations, thereby limiting their applicability to real-world detection tasks.

To address these issues, we construct **RealBench**, a large-scale benchmark designed to reflect practical hybrid text detection settings, which encompasses diverse human–AI collaboration modes, spanning 1,204 text categories (e.g., Llama3_polish_GPT-4o [2, 12], human_polish_Gemini1.5 [13]) and approximately 16.4 million text samples. The dataset covers a wide range of generation types, and its distribution closely reflects real-world usage patterns. We investigate the categorization of hybrid texts and find that, in human–AI collaborative writing, traces of AI involvement tend to be more prominent than human characteristics.

We propose a novel representation learning-based approach, which incorporates structured source modeling, going beyond traditional flat classification. We observe that texts produced by different generation mechanisms naturally exhibit varying degrees of relational similarity. For example, the similarity between texts labeled as "Llama3_polish_GPT-4o" and "Claude3.5_paraphrase_GPT-4o" [14] is higher than their similarity with "human_polish_Gemini1.5", and lowest when compared to purely human-written text. To model this structure, we propose an adaptive algorithm that constructs a **Hierarchical Affinity Tree (HAT)** from the inter-class similarity matrix, with the flexibility to incorporate task-specific structural adjustments. HAT captures intrinsic affinities across arbitrary categories and offers strong interpretability and scalability. Building upon HAT, we develop a **Tree-Structured Contrastive Loss (TSCL)** that explicitly aligns the embedding space with the hierarchical structure. This alignment enhances representation quality.

Moreover, with the rapid emergence of new LLMs, an effective detector should exhibit generalization ability to identify AI involvement from previously unseen LLMs under out-of-distribution (OOD) settings. While many detectors achieve high accuracy on training distributions or under minor domain shifts, their performance degrades under more severe distribution changes [15]. To address this, we propose a retrieval-based few-shot adaptation paradigm that reformulates cross-domain detection as a matching problem under low-resource conditions. The proposed method adjusts the classification decision boundary using limited support samples and demonstrates strong adaptability in the presence of severe distribution shifts, achieving improvements in AUROC under OOD settings, including +15.55 on MAGE-Paraphrase [16], an average of +7.94 on DetectRL [17], and +10.39/+15.01 on the GPT4o-Edited and Llama3.1-Edited subsets of Beemo [18].

In summary, our contributions are threefold: (I) proposing a novel representation learning-based classification paradigm that constructs a Hierarchical Affinity Tree among text categories; (II) constructing a realistic benchmark dataset that simulates human–AI collaborative writing scenarios and conducting a systematic analysis of hybrid texts; (III) providing an effective solution and a promising direction for training-based strategies to detect AI-generated content under OOD settings.

## 2 Related Works

**Detection of Purely-Generated Text.** Transformers and language models have advanced rapidly in recent years, giving rise to a variety of techniques for detecting AI-generated text. Prior work in this area explores multiple paradigms. Watermarking methods [19, 20, 21, 22, 23, 24] embed identifiable patterns into generated content, which can be identified during downstream detection using specialized algorithms. Statistical methods [25, 26, 27, 28, 29, 30, 31] exploit the characteristic probabilistic distributions inherent in the text generation process of language models. These methods employ manually crafted statistical features to differentiate machine-generated text from human-written content. With the growing availability of large-scale datasets, data-driven approaches [6, 32, 33] have achieved notable progress. Ghostbuster [34] extracts token-level features from multiple weak language models, conducts structured feature selection, and uses the selected signals to train a classifier for AI text detection. T5-Sentinel [35] leverages intermediate hidden representations from the T5 model [8] to perform classification. DeTeCtive [36] proposes a supervised detection framework based on stylistic feature learning. To improve the generalization of detectors to emerging models and unseen domains, recent work has explored few-shot learning approaches for detection. OUTFOX [37]

employs in-context learning by incorporating a small number of examples into the prompt, enabling the model to perform both classification and adversarial example generation. UAR [38] leverages a limited set of text samples generated by a specific LLM to quickly localize the model's position in the stylistic space for detection purposes.

**Detection of Hybrid or Collaborative Text.** With the widespread adoption of AI-assisted writing tools, an increasing amount of text is now collaboratively authored by both humans and machines, resulting in "hybrid" content. Several studies [39, 40, 41] have explored sentence-level and boundary-level detection of such human–AI collaborative text. MIXSET [42] reveals the failure of mainstream detectors in reliably identifying hybrid content. APT-Eval [15] shows that existing detectors often suffer from high false positive rates, limited discrimination ability, and strong biases toward older and smaller models when faced with lightly AI-polished text. HART [11] proposes a hierarchical detection framework based on decoupling content and expression, enabling multi-level identification of AI involvement, such as distinguishing between AI-assisted refinement (low-risk) and fully AI-generated text (high-risk). Unlike Hierarchical Text Classification (HTC) [43, 44, 45], where the label hierarchy is predefined and multi-path annotation is explicitly supported, hybrid text detection involves inherently ambiguous and dynamically evolving labels that lack fixed structural definitions. Therefore, the strong structural assumptions of HTC, like rigid tree taxonomies, are poorly suited to the flexible and ambiguous nature of hybrid text categories.

# 3 Method

An overview of the proposed **DETree** framework is provided in Figure 1, which outlines the key components of our method. Section 3.1 presents the motivation and construction of RealBench. Section 3.2 introduces the construction of the Hierarchical Affinity Tree (HAT) based on class-level representation similarities. Section 3.3 formulates the Tree-Structured Contrastive Loss (TSCL), which leverages the learned HAT to guide representation learning. Section 3.4 further describes the implementation details of the inference with DETree.

## 3.1 Motivation and RealBench Construction

In this work, we focus on the detection of human–AI collaborative text. Given an input text sequence $x = \{w_1, w_2, \ldots, w_L\}$, the goal is not only to determine whether LLMs were involved in the generation process, but also to identify the specific form of involvement, such as single LLM generation, multi-LLM collaboration, human-written text edited by LLMs, LLM-generated text revised by a human. Variations in text generation processes can introduce semantic and stylistic signatures that reflect the underlying generation source. For example, the similarity between samples labeled as "Llama3_polish_GPT-4o" [2, 12] and "Claude3.5_paraphrase_GPT-4o" [14] is expected to be higher than their similarity with "DeepSeek-V3" [4], since both involve refinement using GPT-4o. Meanwhile, their similarity to human-written text should be the lowest, as their generation processes are fundamentally different. These interactions give rise to a hierarchical similarity structure among generation sources. However, existing approaches predominantly rely on flat class modeling, failing to capture and exploit this structurally informative signal.

Meanwhile, given that existing datasets focus on purely human-written or AI-generated text, we construct a large-scale hybrid text benchmark, **RealBench**, by aggregating original samples from MAGE [16], M4 [46, 47], TuringBench [48], OUTFOX [37], and RAID [49]. Based on these samples, RealBench introduces a range of hybrid text construction strategies that reflect real-world human–AI collaboration patterns, including paraphrasing, extension, polishing, and translation. For example, a human-written text polished by Gemini1.5 is labeled as "human_polish_Gemini1.5". Additionally, RealBench integrates 11 perturbation-based attack types. See Appendix I for more details.

## 3.2 Hierarchical Affinity Tree Construction

We propose a Hierarchical Affinity Tree Construction Algorithm to formalize and leverage latent relational structures among text categories. Specifically, we fine-tune an encoder $f_\theta(\cdot)$ using supervised contrastive learning [50], treating each category as an independent class. The expected inter-class similarity between any pair $(X, Y)$ is computed as the dot product of their corresponding centroid vectors, where each centroid is computed as the mean embedding across all samples within that

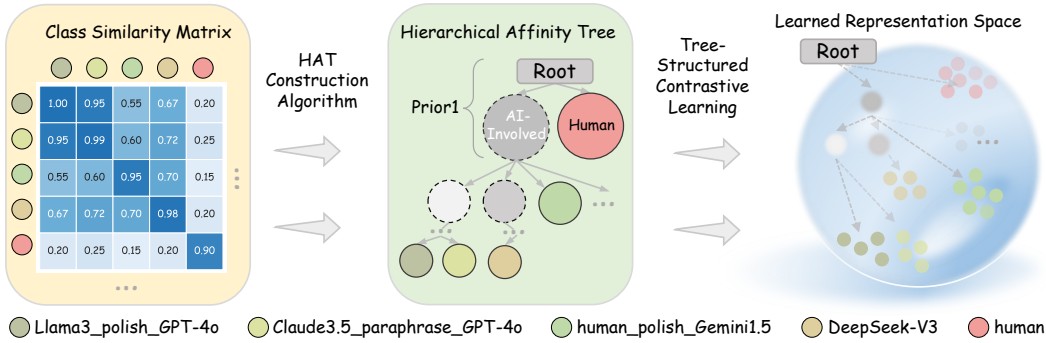

Figure 1: The overall framework of DETree. Class similarity matrix is computed from representations learned via supervised contrastive learning, with each class treated independently. Based on the similarity matrix, Hierarchical Affinity Tree (HAT) is constructed. Guided by the HAT, Tree-Structured Contrastive Loss is introduced to retrain the text encoder, aligning the representation space with the hierarchical relations defined by the HAT.

category, as shown in equation 9. Although the encoder is trained to maximize inter-class separability, it still exhibits latent clustering tendencies among related categories in the embedding space, as shown in Figure 11, suggesting the presence of hierarchical relations.

Motivated by the observed latent structure in the embedding space, we construct Hierarchical Affinity Tree (HAT) to explicitly represent inter-category relationships. In the HAT, leaf nodes correspond to specific categories, while internal nodes capture their associations. The depth of the lowest common ancestor between two nodes reflects the degree of similarity between the corresponding categories.

The initial HAT is constructed based on a class similarity matrix $E \in \mathbb{R}^{N \times N}$. We employ a agglomerative hierarchical clustering algorithm [51] to generate an initial binary tree structure. Since hierarchical clustering enforces a strictly binary tree structure, closely related categories may be unnecessarily split across branches.

In response, we introduce an editable top-down subtree reorganization algorithm. We first predefine three prior heads according to whether hybrid texts belong to the human category, the AI category, or form an independent category, and then use these heads to guide subtree restructuring. See Appendix B for details. For each subtree $x$ under reconstruction, we enumerate all possible partitions of the subtree, assess their clustering quality using the Silhouette Score [52], and select the partition with the highest score. This process is recursively applied until a predefined stopping criterion is met. The overall computational complexity of the algorithm is $\mathcal{O}(N^2 \log N)$, and further implementation details are provided in Appendix C.

### 3.3 Tree-Structured Contrastive Learning

To align the learned representations with the hierarchical structure defined by the HAT, we propose a Tree-Structured Contrastive Loss (TSCL) that explicitly models the hierarchical relations among categories in the representation space. Formally, let $\mathcal{T}$ denote the HAT constructed over the full set of categories. Each category label $y$ corresponds to a leaf node $c$ in $\mathcal{T}$, where $d_c$ denotes its depth. The $i$-th ancestor of node $c$, counting from the bottom (with $i = 0$ corresponding to $c$ itself), is denoted by $f_c^{(i)}$. Starting from $c$, we partition all categories into $d_c$ disjoint sets based on their lowest common ancestor (LCA) with $c$. The $i$-th hierarchical partition set associated with node $c$ is defined as:

$$H_c^{(i)} = \left\{ x \,\middle|\, x \in \text{leaf}(f_c^{(i)}) \setminus \text{leaf}(f_c^{(i-1)}) \right\}, \tag{1}$$

where leaf$(x)$ denotes the set of leaf nodes under the subtree rooted at node $x$. Thus, $H_c^{(i)}$ contains all categories whose nearest common ancestor with $c$ is exactly $f_c^{(i)}$. By definition, $H_c^{(0)} = \{c\}$.

Under the tree $\mathcal{T}$, categories sharing a closer common ancestor with $c$ are expected to exhibit higher similarity and lie closer in the embedding space. To formalize this intuition, we introduce the following hierarchical similarity constraint.

**Theorem 3.1** *[Hierarchical Similarity Constraint] If in the tree $\mathcal{T}$, for any leaf class X corresponding to node c, the following holds:*

$$\mathbb{E}\big[\text{sim}(X,Y)\big] > \mathbb{E}\big[\text{sim}(X,Z)\big], \quad \forall 0 \leq i < j \leq d_c, \ Y \in H_c^{(i)}, \ Z \in H_c^{(j)}. \tag{2}$$

*Then, for any leaf classes X, Y, Z, the following inequalities holds:*

$$\mathbb{E}\big[\text{sim}(X,Y)\big] > \mathbb{E}\big[\text{sim}(X,Z)\big], \quad \textit{if } d_{\text{LCA}(X,Y)} > d_{\text{LCA}(X,Z)}. \tag{3}$$

*Where $\mathbb{E}[\text{sim}(\cdot, \cdot)]$ denotes the expected similarity between categories.*

By Theorem 3.1, inequalities 2 and 3 are equivalent. The proof is provided in Appendix D.

To encourage the embedding space to better align with the hierarchical structure modeled by the HAT, we aim to optimize the inequalities 3 during training, such that categories that are closer in the HAT structure are mapped to more similar representations, while those farther apart are pushed away.

However, directly optimizing inequalities 3 is challenging. Therefore, based on the equivalence in Theorem 3.1, we transform the objective to the more tractable inequalities 2. Accordingly, we adopt the following training objective:

$$\max_{\theta} \ \mathbb{E}_{x \sim \mathcal{D}}\big[\mathcal{G}(x; \theta)\big], \tag{4}$$

$$\mathcal{G}(x;\theta) = \sum_{0 \leq i < j \leq d_c} \Big( \mathbb{E}_{y \sim \mathcal{U}(H_c^{(i)})}\big[\text{sim}\big(f_\theta(x), f_\theta(y)\big)\big] - \mathbb{E}_{z \sim \mathcal{U}(H_c^{(j)})}\big[\text{sim}\big(f_\theta(x), f_\theta(z)\big)\big] \Big). \tag{5}$$

Here in equation 4, $\mathcal{D}$ denotes the distribution of text samples $x$, $c$ denote its corresponding leaf node in the HAT, and $d_c$ its depth. In equation 5, the function $f_\theta(x) \in \mathbb{R}^d$ denotes the embedding of $x$ computed by the encoder $f_\theta$. The sets $H_c^{(i)}$ are defined as in equation 1, and $\mathcal{U}(H_c^{(i)})$ denotes the distribution over $H_c^{(i)}$. The similarity function $\text{sim}(\cdot, \cdot)$ measures the pairwise similarity between embeddings. The objective $\mathcal{G}(x; \theta)$ computes the expected similarity gap between all hierarchical pairs $(i, j)$ of categories associated with node $c$.

Motivated by contrastive learning, we reformulate the hierarchical similarity constraint into a structured positive–negative sampling scheme tailored to the tree hierarchy. For each hierarchical level $i$ $(0 \leq i < d_c)$, we define the positive set $P_i = H_c^{(i)}$ and the negative set $N_i = \bigcup_{j=i+1}^{d_c} H_c^{(j)}$. The complete Tree-Structured Contrastive Loss (TSCL) is defined as:

$$\mathcal{L}_{\text{TSCL}}(x; \theta) = \frac{1}{d_c} \sum_{i=0}^{d_c-1} \mathcal{L}_c^{(i)}(x; \theta), \tag{6}$$

where each hierarchical contrastive loss $\mathcal{L}_c^{(i)}(x; \theta)$ is computed as:

$$\mathcal{L}_c^{(i)}(x;\theta) = -\log \frac{\exp\Big(\frac{1}{|P_i|}\sum_{k \in P_i} \text{sim}\big(f_\theta(x), f_\theta(k)\big)/\tau\Big)}{\exp\Big(\frac{1}{|P_i|}\sum_{k \in P_i} \text{sim}\big(f_\theta(x), f_\theta(k)\big)/\tau\Big) + \sum_{k \in N_i} \exp\big(\text{sim}\big(f_\theta(x), f_\theta(k)\big)/\tau\big)}. \tag{7}$$

Where $\tau > 0$ is a temperature parameter, and $\text{sim}(\cdot, \cdot)$ denotes the similarity function (e.g., cosine similarity) between embeddings. This formulation reflects a hierarchy-aware similarity structure, encouraging representations to be increasingly similar to categories with closer shared ancestors in the tree, and more dissimilar to those located further apart in the hierarchy.

In practice, due to the large number of categories $\mathcal{C}$, a single training mini-batch $\mathcal{B}$ may not contain sufficient instances to fully cover all categories. Consequently, We introduce the Virtual Class Prototype (VCP) mechanism: for each class $c$, a learnable prototype vector $\mathbf{v}_c \in \mathbb{R}^d$ is introduced to participate in contrastive learning as a persistent anchor, without incurring additional memory overhead. The entire TSCL training process supports parallel computation and incurs similar runtime to standard contrastive learning.

### 3.4 Inference with DETree

**K-Means Based Database Compression.** In the task of AI-generated text detection, samples typically originate from multiple domains. Even when the source is the same, substantial variation in domain may lead to multi-cluster distributions (see Figure 2). In such scenarios, K-Nearest Neighbors (KNN) classification methods [53] naturally adapt to complex distributions and allow flexible modification of the retrieval database according to task-specific requirements. However, their computational and storage costs grow linearly with the size of the database. Moreover, when class sizes are imbalanced or spatial distributions are non-uniform, decision boundaries tend to become distorted. To address this, we propose a database compression method aimed at reducing the number of retrieval candidates while maintaining a balanced class representation. Specifically, for each class, we apply K-means [54] clustering to partition its instances into $K$ clusters. The normalized direction of the mean vector of each cluster is then used as its representative, resulting in a compact class representation. More details are provided in Appendix E.

**Retrieval-Based Few-Shot Adaptation.** Distributional discrepancies arise between datasets due to differences in text sources, prompt types, and generation model configurations, reflecting domain shifts. Conventional binary classification methods tend to rely on superficial features, which limits their ability to generalize across domains. As a structured source modeling approach, DETree relies on a retrieval database constructed from training data. However, when this data lacks sufficient coverage, the resulting database fails to generalize to target domains, leading to performance bottlenecks under out-of-distribution (OOD) scenarios. To address this, we incorporate a small number of target-domain samples to rebuild the retrieval database, enabling more accurate decision boundaries in the embedding space and significantly improving detection performance under domain shift.

## 4 Experiments

### 4.1 Experimental Setup

**Dataset.** We conduct in-distribution supervised experiments on the MAGE [16], M4 [46, 47], TuringBench [48], OUTFOX [37], and RAID [49] datasets. Given substantial differences across benchmarks in terms of generation models, human text sources, domain distributions, and prompting configurations, we treat unseen benchmarks (DetectRL [17], Beemo [18], HART [11], and the OOD setting of MAGE) as out-of-distribution test sets for evaluating model generalization. Detailed dataset specifications are provided in Appendix H and I.

**Comparison Methods and Metrics.** We evaluate zero-shot methods including Fast-DetectGPT [28] (shortly Fast-Detect), Binoculars [29], and Glimpse [55]; hybrid text classification via HART [11]; few-shot classification via UAR [38]; and data-driven methods including SCL [56], MAGE [16], T5-Sentinel [35], DeTeCTive [36], and RADAR [9]. Since large language models in real-world scenarios are often black-box and inaccessible, we exclude watermarking methods. Evaluation metrics include F1, AvgRec (mean recall of human and machine text), and AUC-ROC. We additionally report TPR@5% FPR to assess detection performance under low false positive constraints.

**Implementation Details.** We fine-tune RoBERTa-large [57] using LoRA [58], with AdamW [59], cosine annealing, a 3e-5 initial learning rate, and 2000 step linear warm-up. Training runs for 10 epochs on 8×RTX 4090 GPUs with a batch size of 64 and a maximum input length of 512 tokens. For inference, we use Faiss-GPU [60] for efficient K-means and K-Nearest Neighbors. The representation layer is selected from layers 17–19, and the number of neighbors $k$ is set to 5 or 50 based on validation performance. To compute class probabilities, we follow DINOv2 [61] and replace majority voting with similarity-weighted scoring over the retrieved neighbors. The contrastive-learning temperature $\tau$ is set to 0.07.

### 4.2 Results

We explore our method across five key dimensions: (I) HAT Analysis: effectiveness of TSCL and interpretability of the HAT structure; (II) Supervised Detection: model performance on in-distribution binary classification tasks; (III) Out-of-Distribution Generalization: model performance on binary

Table 1: Comparison of supervised detection performance. *w/ per dataset* denotes DETree trained individually on each dataset. *w/ prior1/2/3* indicates DETree trained on RealBench under prior assumptions prior 1/2/3 (see Figure 7). *w/o TSCL* refers to supervised contrastive learning performed independently for each class, without the TSCL. All baseline methods, except Binoculars, are trained in a fully supervised manner on the corresponding datasets. Best results are **bolded**.

| Method | MAGE | | M4-monolingual | | M4-multilingual | | TuringBench | | Average | |
|---|---|---|---|---|---|---|---|---|---|---|
| | AvgRec | F1 | AvgRec | F1 | AvgRec | F1 | AvgRec | F1 | AvgRec | F1 |
| SCL | 90.59 | 89.83 | 91.92 | 91.21 | 86.27 | 84.75 | 99.46 | 99.22 | 92.06 | 91.25 |
| RoBERTa-Base | 87.30 | 88.37 | 88.70 | 88.44 | 80.01 | 84.44 | 99.59 | 99.29 | 88.90 | 90.14 |
| MAGE | 90.53 | 89.76 | 80.99 | 81.42 | 84.68 | 83.00 | 99.40 | 98.95 | 88.90 | 88.28 |
| T5-Sentinel | 93.49 | 93.30 | 84.01 | 81.08 | 76.21 | 68.99 | 99.39 | 97.43 | 88.28 | 85.20 |
| Binoculars | 64.96 | 70.58 | 89.89 | 89.89 | 80.63 | 82.43 | 51.24 | 9.98 | 71.68 | 63.22 |
| DeTeCTive | 96.15 | 96.16 | 98.44 | 98.38 | 93.42 | 93.05 | 99.74 | 99.35 | 96.94 | 96.74 |
| **DETree (Ours)** | | | | | | | | | | |
| *w/ per dataset* | **96.97** | **96.98** | 98.96 | 98.92 | 93.62 | 93.35 | 99.62 | **99.39** | 97.29 | 97.16 |
| *w/ prior1* | 96.87 | 96.96 | **99.86** | **99.85** | 95.05 | 94.85 | **99.74** | 99.32 | **97.88** | **97.75** |
| *w/ prior2* | 95.45 | 95.63 | 98.17 | 98.02 | 91.77 | 91.72 | 99.29 | 98.75 | 96.17 | 96.03 |
| *w/ prior3* | 96.15 | 96.28 | 99.52 | 99.47 | **95.53** | **95.33** | 99.52 | 98.88 | 97.68 | 97.49 |
| *w/o TSCL* | 95.65 | 95.82 | 98.84 | 98.73 | 92.11 | 91.89 | 99.39 | 98.83 | 96.50 | 96.32 |

classification under OOD settings; (IV) Hybrid Text Detection: the ability to identify varying forms of AI involvement in collaborative text generation; (V) Practical Robustness and Deployability: model performance under real-world constraints, including adversarial perturbations, database compression, and limited category coverage.

**HAT Analysis.** A total of 99.32% of triplets satisfy the constraint defined in Theorem 3.1 after training with TSCL, confirming the effectiveness of the proposed loss. Appendix J provides an intuitive visualization of the intermediate clustering structures in the HAT construction process. We observe that HAT is capable of autonomously capturing correlations among text sources: within hybrid texts, samples from the same model using the same transformation strategy (e.g., paraphrasing, extension, translation) tend to cluster together. Within such clusters, texts based on the same model family exhibit further aggregation, forming terminal leaf nodes. Moreover, some non-family models (e.g., LLaMA [62] and GLM-130B [63], FLAN-T5 [64] and T0 [65]) are also grouped together, potentially reflecting similarities in their pretraining data or pretraining model (T5) [8]. Models from different datasets cluster separately, suggesting distributional discrepancies.

**Supervised Detection.** As shown in Table 1, our method demonstrates superior performance in supervised settings. Training with the Tree-Structured Contrastive Loss (TSCL) leads to the best performance under prior1, while removing TSCL results in a noticeable performance drop. We additionally investigate the impact of different prior assumptions on model performance. Among the tested priors, prior1 yields the best results, followed by prior3, while prior2 performs the worst. This outcome aligns with the intrinsic nature of hybrid text: categorizing human–AI collaborative texts as AI-involved proves to be more appropriate. Once AI is introduced into the generation process, even minimally, its stylistic signals tend to dominate and are often more salient than human-authored traits. This observation is further supported by the dimensionality reduction visualizations in Figure 2.

**Out-of-Distribution Generalization.** As shown in Table 2, under the zero-shot setting, evaluation metrics vary across different database configurations, indicating distributional discrepancies among the datasets. RAID and RealBench yield relatively strong results. The model performs well on MAGE's Unseen setting and the DetectRL scenarios but shows weaker performance on Beemo, primarily due to mismatches between the retrieval database and the test distribution under OOD scenarios. To mitigate this issue, we incorporate a small number of target-domain samples to adjust the decision boundaries. Under the few-shot setting, the inclusion of such samples improves classification performance, particularly on datasets where zero-shot accuracy is low. Beemo's No Attack setting remains challenging; however, detection performance improves substantially on texts that have undergone further LLM editing. This may be because the additional editing amplifies LLM-specific stylistic features, resulting in a more concentrated distribution that is easier to detect.

Table 2: OOD evaluation results, measured by AUROC. All methods are evaluated in a zero-shot setting without any training. DETree (database) denotes evaluations where different datasets are used as the retrieval database. DETree (few-shot) and UAR denote few-shot evaluations, where 5 or 10 samples are randomly drawn for each source type from the target validation or test set. Each experiment is repeated five times, and the average performance is reported. Best results are **bolded**.

| Dataset | Setting | Fast-Detect | Binoculars | MAGE | RADAR | DETree (database) | | | | UAR | DETree (few-shot) |
|---|---|---|---|---|---|---|---|---|---|---|---|
| | | | | | | RealBench | MAGE | M4 | RAID | 5/10 shot | 5/10 shot |
| MAGE | Unseen | 80.76 | 96.84 | 95.20 | 87.37 | **99.13** | 97.78 | 94.92 | 98.98 | 72.07 / 74.22 | 99.58 / **99.81** |
| | Paraphrase | 53.34 | 75.87 | 83.35 | 71.11 | 90.80 | 88.29 | 88.36 | **92.66** | 59.54 / 58.18 | **98.90** / 98.77 |
| DetectRL | Multi-Domain | 58.52 | 83.95 | 86.67 | 92.15 | 98.74 | 98.15 | 98.09 | **98.94** | 73.84 / 75.39 | 99.62 / **99.88** |
| | Multi-LLM | 59.58 | 83.30 | 86.26 | 91.86 | 98.62 | 98.11 | 98.08 | **98.92** | 69.97 / 73.92 | 99.52 / **99.87** |
| | Multi-Attack | 60.70 | 85.05 | 88.52 | 91.80 | 98.88 | 98.21 | 98.15 | **99.00** | 73.49 / 69.91 | 99.65 / **99.87** |
| Beemo | No-Attack | 75.46 | **83.90** | 73.72 | 51.41 | 81.59 | 75.76 | 79.09 | 79.99 | 62.73 / 64.40 | 79.32 / **81.64** |
| | GPT4o-Edited | 62.64 | 78.15 | 67.79 | 60.16 | 82.07 | 75.01 | 77.44 | **83.79** | 65.52 / 66.36 | 88.23 / **88.54** |
| | Llama3.1-Edited | 65.43 | 79.90 | 65.37 | 62.65 | **87.16** | 75.02 | 81.74 | 83.04 | 63.61 / 67.57 | 93.81 / **94.91** |

**Hybrid Text Detection.** Table 3 reports the results on the three-level AI risk detection task proposed by HART, with details in Appendix H. Level-1 identifies whether the text involves any AI participation, Level-2 distinguishes whether the base content is human-written or AI-generated, and Level-3 detects whether the text is solely produced by a single AI model. Results show that, given a well-defined database split corresponding to each task, DETree consistently achieves effective discrimination across all three levels, demonstrating the fine-grained discriminative power of the learned representations.

Table 3: AUROC and TPR@5%FPR (shortly TPR5%) results for the AI risk detection tasks on HART. The column labeled **ALL** aggregates results across multiple English subdomains, including Essay, ArXiv, Writing, and News. DETree is trained on RealBench under different prior assumptions and evaluated using the HART development set as the retrieval database. HART(Fast-Detect / Binoculars) employs the development set to fit its two-dimensional binary classifier. Best results are **bolded**.

| Detector | Level-3 | | | | Level-2 | | | | Level-1 | | | |
|---|---|---|---|---|---|---|---|---|---|---|---|---|
| | Essay | ArXiv | Writing | ALL(TPR5%) | Essay | ArXiv | Writing | ALL(TPR5%) | Essay | ArXiv | Writing | ALL(TPR5%) |
| RADAR | 0.692 | 0.849 | 0.647 | 0.728 (14%) | 0.566 | 0.814 | 0.630 | 0.687 (10%) | 0.705 | 0.857 | 0.700 | 0.758 (20%) |
| Log-Perplexity | 0.868 | 0.850 | 0.811 | 0.799 (33%) | 0.364 | 0.485 | 0.438 | 0.473 (11%) | 0.769 | 0.530 | 0.625 | 0.576 (6%) |
| Log-Rank | 0.867 | 0.874 | 0.813 | 0.814 (39%) | 0.380 | 0.460 | 0.441 | 0.465 (11%) | 0.739 | 0.542 | 0.611 | 0.573 (8%) |
| LRR | 0.835 | 0.909 | 0.797 | 0.840 (50%) | 0.560 | 0.616 | 0.551 | 0.573 (25%) | 0.616 | 0.576 | 0.558 | 0.568 (19%) |
| Glimpse | 0.929 | 0.869 | 0.819 | 0.849 (58%) | 0.754 | 0.737 | 0.625 | 0.676 (30%) | 0.878 | 0.719 | 0.618 | 0.688 (22%) |
| Fast-Detect | 0.883 | 0.877 | 0.840 | 0.862 (60%) | 0.734 | 0.718 | 0.692 | 0.711 (47%) | 0.877 | 0.769 | 0.740 | 0.778 (55%) |
| Binoculars | 0.897 | 0.882 | 0.847 | 0.870 (62%) | 0.735 | 0.715 | 0.693 | 0.711 (44%) | 0.879 | 0.769 | 0.740 | 0.780 (55%) |
| HART(Fast-Detect) | 0.864 | 0.896 | 0.890 | 0.876 (61%) | 0.785 | 0.915 | 0.890 | 0.855 (59%) | 0.907 | 0.849 | 0.836 | 0.843 (59%) |
| HART(Binoculars) | 0.854 | 0.904 | 0.905 | 0.883 (61%) | 0.746 | 0.913 | 0.895 | 0.848 (32%) | 0.900 | 0.840 | 0.828 | 0.838 (58%) |
| **DETree (Ours)** | | | | | | | | | | | | |
| *w/ prior1* | **0.984** | **0.991** | **0.983** | **0.988** (95.3%) | **0.994** | **0.998** | **0.990** | **0.992** (98.5%) | 1.00 | **0.999** | 0.996 | **0.998** (99.5%) |
| *w/ prior2* | 0.976 | 0.986 | 0.959 | 0.972 (88.7%) | 0.981 | 0.992 | 0.973 | 0.982 (93.6%) | 0.989 | 0.997 | 0.994 | 0.994 (98.9%) |
| *w/ prior3* | 0.983 | **0.991** | 0.973 | 0.979 (92.2%) | 0.993 | 0.997 | 0.988 | **0.992** (96.8%) | 0.999 | 0.998 | **0.998** | **0.998** (99.7%) |

To further investigate the interrelations among various types of hybrid texts and evaluate DETree's ability to distinguish them, we perform a comprehensive analysis on the HART dataset, as shown in Figure 2, Figure 3 and Figure 8.

Based on the dimensionality-reduced representations, although the training objective encourages clustering of all human-written texts, the model still separates human texts from different domains. This indicates that semantic information implicitly influences the modeling of text provenance; even without explicit supervision, the model is able to autonomously capture and leverage such distinctions.

Except for the machine humanized human category, the other five types of texts exhibit well-formed clusters in the embedding space and are clearly separable in binary classification tasks. An interesting observation is that although the training data does not include samples humanized by commercial tools, the model successfully identifies such texts. Machine humanized human texts tend to be indistinguishable from those based on machine-generated base texts, but are distinguishable when the base text is human-written. Human edits may preserve machine-originated styles or reflect refinement patterns resembling tools or LLMs. This suggests that human editing is diverse but does not fundamentally alter the characteristics of the base text.

In summary, **DETree** can detects AI-involved content, identifies the provenance of the base text (human or machine), and identifies whether and how it has been humanized by specific methods. We provide further analysis of hybrid texts on HART, including multilingual settings, visualizations, and

HAT construction in Appendix F; we additionally present the detection of hybrid texts involving three authors in Appendix G.

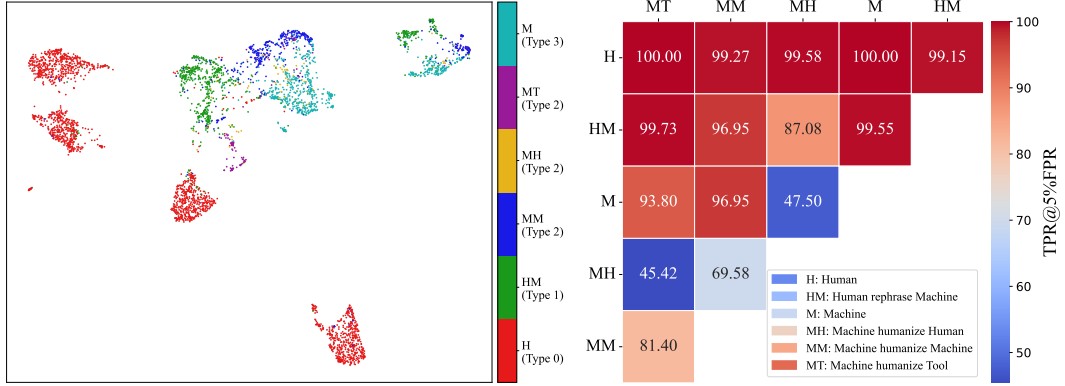

Figure 2: UMAP-based unsupervised visualization of the HART dataset in the representation space of the DETree. Colors indicate different text source types, with type indices corresponding to the hierarchical categorization defined in HART. Abbreviations of category names are defined in Figure 3.

Figure 3: Pairwise binary classification performance heatmap on the HART dataset, evaluated by TPR@5%FPR. Rows and columns represent the two text types involved in each detection pair. In the legend, MT denotes machine-generated text humanized by tools.

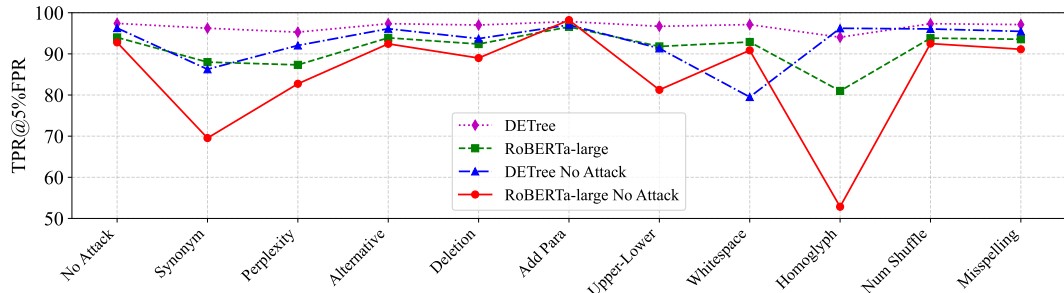

Figure 4: Robustness evaluation of different detection methods on the attack-augmented MAGE test set, measured by TPR@5%FPR. The x-axis denotes perturbation types. "No Attack" indicates that the model was trained without adversarial samples, while others were trained with all attack types. DETree is evaluated using a compressed 10K-sample version of RealBench as the database.

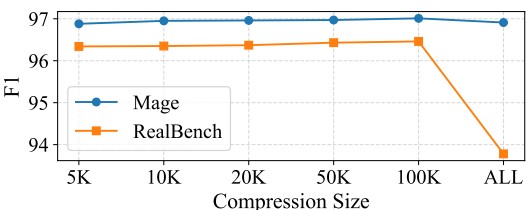

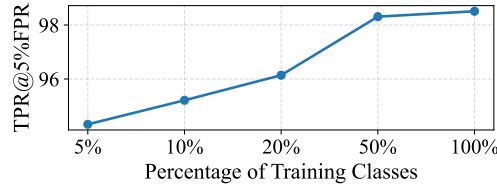

Figure 5: Detection performance (F1) on the MAGE test set using MAGE (~500K) and RealBench (~12M) as the database under different compression sizes. "ALL" indicates no compression.

Figure 6: Detection performance on the MAGE test set (TPR@5%FPR) when training on RealBench with randomly selected subsets of classes at different proportions.

**Practical Robustness and Deployability.** As shown in Figure 4, incorporating adversarial samples during training improves model performance under various perturbations. DETree demonstrates stable performance across all attack types when trained with such examples, while RoBERTa-large still suffers substantial degradation under Synonym, Perplexity, and Homoglyph attacks. Without adversarial training, RoBERTa-large becomes highly vulnerable to perturbations. In comparison,

DETree maintains strong performance even without adversarial exposure, consistently outperforming the adversarially trained RoBERTa-large in most attack scenarios.

As shown in Figure 5, we investigate the impact of K-means based database compression on detection performance. On the RealBench dataset, compressing the full set of training samples into 10K learnable representatives not only reduces storage and computational overhead, but also leads to a slight improvement in detection accuracy. To further assess the influence of category coverage on model performance, we conduct experiments under varying proportions of training categories. As illustrated in Figure 6, increasing the number of training categories enhances model performance. Notably, when trained on only 10% of the full category set, the performance drop remains within 3%.

## 5 Conclusion

In this study, we propose **DETree**, a novel representation learning-based detection framework designed to address the challenges of identifying human-AI hybrid text in complex real-world scenarios. We construct **RealBench**, a large-scale benchmark dataset that encompasses diverse modes of human–AI collaborative writing. By explicitly modeling the hierarchical relationships among text sources, DETree reveals that hybrid texts generated through human–AI collaboration exhibit stronger AI traces than human characteristics. Extensive experiments demonstrate that our method achieves state-of-the-art performance across multiple benchmark tasks and maintains strong generalization capabilities under low-supervision conditions and severe distribution shifts.

## 6 Limitations

While our method demonstrates strong adaptability to real-world detection scenarios, it still exhibits minor limitations in certain aspects. We have not explored adversarial evasion via model fine-tuning intended to bypass detection. While this work focuses on hybrid texts involving three authors, extending the analysis to scenarios with more collaborators remains an interesting direction for future investigation. Furthermore, when encountering entirely unseen and rare domains, the model still requires a small number of in-domain samples to effectively adjust the decision boundaries.

## Acknowledgments

The work was supported by National Key Research and Development Program of China (No. 2022YFB2702500), the National Natural Science Foundation of China (No. 62206265, 62076231).

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

## A  Broader Impacts

This study aims to improve the detection of AI-involved text, particularly by enabling more fine-grained distinction in human–AI hybrid text scenarios. We believe that this technology can play a positive role in enhancing text transparency and safeguarding academic integrity and media credibility. However, we also recognize that its application may carry certain potential risks. For example, if used for excessive surveillance or content censorship, it could negatively impact legitimate information expression and freedom of speech. On the other hand, stronger detection capabilities may drive the development of generative models that are designed to evade detection. Therefore, when applying this method, we recommend taking the specific usage context into account, paying attention to the boundaries of its application, and promoting its development within an open and transparent framework to ensure that it produces positive social outcomes.

## B  Prior

To obtain a prior structure suited to the task characteristics, we consider all possible similarities between hybrid texts and other generation types, and propose three prior assumptions: prior 1 assumes that hybrid texts are closer to AI-generated content; prior 2 assumes that human features are more prominent; prior 3 treats hybrid texts as a distinct category, separate from both AI and human texts. The three priors are illustrated in Figure 7.

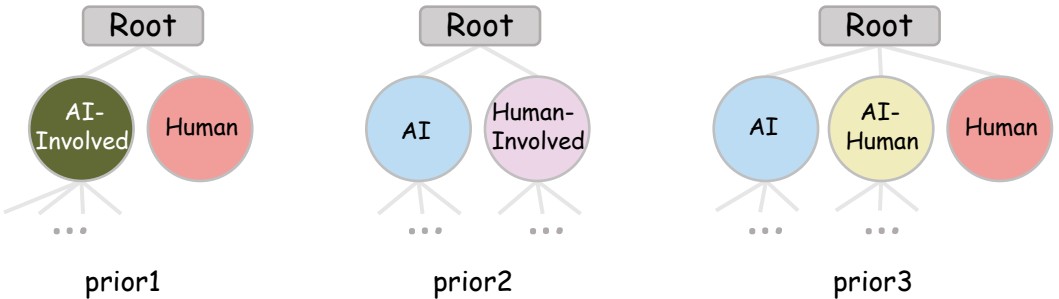

Figure 7: Illustration of three prior assumptions used during training, each defining a different hierarchical placement for human–AI collaborative texts. *prior 1* assigns them as a subtree under AI-generated texts; *prior 2* places them under human-written texts; and *prior 3* treats them as an independent branch parallel to both AI and human texts.

## C  HAT Construction

### C.1  Preliminaries

**Agglomerative Hierarchical clustering**   Agglomerative hierarchical clustering [51] is a bottom-up unsupervised method that constructs a binary tree to capture the hierarchical relationships among samples. Starting with each sample as an individual cluster, the algorithm iteratively merges the two most similar clusters based on a predefined distance metric until a single cluster remains. At each merge step, a new parent node is created to link the two child clusters, and the corresponding merge similarity (denoted as the **merge_score**) is recorded. Common inter-cluster distance metrics include minimum, maximum, and average linkage; we adopt average linkage to encourage a more balanced tree structure.

**Silhouette Score**   Silhouette Score [52] is a standard metric for evaluating clustering quality by assessing the balance between intra-cluster cohesion and inter-cluster separation. For each sample, let $a$ denote the average distance to other points within the same cluster, and $b$ the average distance to points in the nearest neighboring cluster. The silhouette score is defined as $s = \frac{b-a}{\max(a,b)}$, with values ranging from $-1$ to $1$. Higher scores indicate that the sample is well-matched to its own cluster and well-separated from others.

## C.2 Similarity Matrix

To capture inter-class similarity, we first apply supervised contrastive learning by treating each class independently. Specifically, for a given sample $x$, the contrastive loss is defined as:

$$\mathcal{L}(x;\theta) = -\log \frac{\exp\left(\frac{1}{|P|}\sum_{k\in P}\cos(f_\theta(x), f_\theta(k))/\tau\right)}{\exp\left(\frac{1}{|P|}\sum_{k\in P}\cos(f_\theta(x), f_\theta(k))/\tau\right) + \sum_{k\in N}\exp\left(\cos(f_\theta(x), f_\theta(k))/\tau\right)} \quad (8)$$

where $f_\theta(\cdot)$ is a parameterized encoder, $\cos(\cdot,\cdot)$ denotes cosine similarity, $\tau$ is a temperature parameter, $P$ is the set of positive samples sharing the same label as $x$, and $N$ is the set of negatives from other classes. This objective encourages intraclass compactness and interclass separation in the representation space.

We use the trained encoder $f_\theta$ to compute the expected similarity between any pair of classes $(X, Y)$ based on their sample representations

$$\mathbb{E}[\text{sim}(X,Y)] = \frac{1}{N \times M}\sum_{i=1}^{N}\sum_{j=1}^{M} f_\theta(x_i)^\top f_\theta(y_j) = \left(\frac{1}{N}\sum_{i=1}^{N}f_\theta(x_i)\right)^\top \left(\frac{1}{M}\sum_{j=1}^{M}f_\theta(y_j)\right) \quad (9)$$

where class $X$ contains $N$ samples $x_i$ and class $Y$ contains $M$ samples $y_j$. As shown in equation 9, the expected similarity can be simplified as the inner product between the mean embeddings of the two classes.

---

**Algorithm 1** Hierarchical Affinity Tree Construction Algorithm

---

1: **Input:**
2: similarity matrix $S \in \mathbb{R}^{N \times N}$,end score $s$
3: **Output:**
4: Hierarchical Affinity Tree $\mathcal{T}$
5: **Definitions:**
6: `SilhouetteScore`: Clustering quality metric. `Partition`: Divide the subtree of $node$ into subgroups whose root merge_score is bounded below by $\tau$. `AgglomerativeClustering`: Hierarchical clustering method. `node.merge_score`: Subtree similarity at merge.
7: **Step1 Hierarchical Clustering** :
8: Build dendrogram $\mathcal{T}_{hc} \leftarrow$ AgglomerativeClustering$(S)$
9: **Step2 Tree Reconstruction** :
10: **function** RECONSTRUCTION$(node)$
11:     $\mathcal{T} \leftarrow node$
12:     **if** $node.is\_leaf$ **or** $node.merge\_score \geq s$ **then**
13:         **return** $\mathcal{T}$
14:     **end if**
15:     $\tau \leftarrow$ set of merge_scores from all descendant nodes of $node$.
16:     Initialize tracker $\tau^* \leftarrow \emptyset$, $score^* \leftarrow -\infty$
17:     **for** each $\tau' \in \tau$ **do**
18:         Compute the partition $C \leftarrow$ Partition$(node, \tau')$
19:         $score \leftarrow$ SilhouetteScore$(C, S)$
20:         **if** $score > score^*$ **then**
21:             $\tau^* \leftarrow \tau'$, $score^* \leftarrow score$
22:         **end if**
23:     **end for**
24:     $children \leftarrow$ Partition$(node, \tau^*)$
25:     **for** each $child \in children$ **do**
26:         $subtree \leftarrow$ Reconstruction$(child)$
27:         $\mathcal{T}.add\_edge(node, subtree)$
28:     **end for**
29:     **return** $\mathcal{T}$
30: **end function**
31: $\mathcal{T} \leftarrow$ Reconstruction$(\mathcal{T}_{hc}.root, 0)$

---

## C.3 HAT Construction Algorithm

**Algorithm Details**   We first extract the set of categories to be merged within each prior-defined subtree based on the task-specific prior knowledge, along with their corresponding similarity matrix. Using this matrix, we then construct an initial binary tree via Agglomerative Clustering, where each internal node stores the similarity score at which its child subtrees were merged (denoted as $merge\_score$). Since binary tree structures may forcefully separate classes that should reside at the same hierarchical level, we recursively reconstruct the tree in a top-down manner. A stopping threshold $s$ is introduced to prevent further partitioning of nodes with high internal similarity.

Specifically, for each node, we collect the merge_scores of all its descendant nodes as a set of candidate thresholds. For each candidate threshold $\tau$, we extract all subtrees within the current subtree whose root merge_scores are exactly no less than $\tau$, and treat them as partition. We then compute the Silhouette Score for each partition and select the one with the highest score as the optimal structure. The node is then divided into multiple substructures accordingly, and the reconstruction process is applied recursively to each child. This results in a multiway tree structure, forming the final Hierarchical Affinity Tree (HAT). The detailed algorithm can be found in Algorithm 1.

**Time Complexity of HAT Construction**   For the hierarchical clustering step, heap-based optimization reduces the time complexity to $O(N^2 \log N)$. The top-down reconstruction process has a complexity of $O(\text{max\_dep} \times N^2)$, where max_dep denotes the maximum depth of the constructed HAT tree and $N$ is the number of categories. Since the resulting tree is typically balanced, max_dep can be regarded as a constant or at most $O(\log N)$. Therefore, the overall construction complexity is $O(N^2 \log N)$.

## C.4 HAT Depth Ablation

We control the granularity of HAT construction with a silhouette-based termination threshold $\tau$: a smaller threshold encourages further node splitting and thus deeper trees. To quantify its impact, we sweep $\tau \in \{0.30, 0.25, 0.20\}$, which in our implementation correspond to maximum subtree depths of 3, 6, and 9, respectively. We train DETree on RealBench and evaluate on MAGE.

Experimental results in Table 4 indicate that DETree is robust across different depths. When the tree is too shallow, fine-grained category structure is lost and accuracy drops; increasing depth improves performance and peaks around a maximum depth of 6. Pushing the depth further brings only a slight decline. Unless otherwise stated, the main paper reports results using a maximum depth of 6 (i.e., $\tau$=0.25).

Table 4: Effect of maximum HAT depth on DETree performance.

| Max depth | 3 | 6 | 9 |
|---|---|---|---|
| F1 | 96.48 | 96.96 | 96.84 |
| AvgRecall | 96.38 | 96.87 | 96.76 |

# D  Proof of Hierarchical Similarity Constraint

In this section, we prove Theorem 3.1.

## D.1  Restatement of the Theorem

**Theorem (Hierarchical Similarity Constraint)** Let $\mathcal{T}$ be a tree, and let $X, Y, Z$ be arbitrary leaf classes (corresponding to leaf nodes). Define $d_{\text{LCA}(X,Y)}$ as the depth of the lowest common ancestor of $X$ and $Y$ (assume the root node has depth 0, and a larger depth value means closer to the leaf nodes). Then the following two inequalities are equivalent:

1. For any leaf class $X$ (corresponding node $c$),

$$\mathbb{E}\big[\text{sim}(X,Y)\big] > \mathbb{E}\big[\text{sim}(X,Z)\big], \quad \forall 0 \leq i < j \leq d_c, \ Y \in H_c^{(i)}, \ Z \in H_c^{(j)}. \tag{10}$$

where $d_c$ is the depth of $c$, and $H_c^{(i)}$ is the $i$-th hierarchical partition set of $c$ (definition see Section 3.3).

2. For any leaf classes $X, Y, Z$,

$$\mathbb{E}\big[\text{sim}(X, Y)\big] > \mathbb{E}\big[\text{sim}(X, Z)\big], \quad \text{if } d_{\text{LCA}(X,Y)} > d_{\text{LCA}(X,Z)}. \tag{11}$$

## D.2 Proof

To prove that inequalities 10 and inequalities 11 are equivalent, it is necessary to prove the two-way implication: inequalities $10 \Rightarrow$ inequalities 11 and inequalities $11 \Rightarrow$ inequalities 10.

Based on the tree $\mathcal{T}$, for any fixed leaf node $c$, we denote its ancestor sequence $\{f_c^{(i)} \mid i = 0, 1, \ldots, d_c\}$ satisfying $f_c^{(0)} = c$ (depth $d_c$), $f_c^{(1)}$ is its parent node (depth $d_c - 1$), and so on, $f_c^{(d_c)}$ is the root node (depth 0).

### D.2.1 Part I: inequalities $10 \Rightarrow$ inequalities 11

Assume inequalities 10 holds. It is required to prove that for any leaf classes $X, Y, Z$, if $d_{\text{LCA}(X,Y)} > d_{\text{LCA}(X,Z)}$, then $\mathbb{E}\big[\text{sim}(X, Y)\big] > \mathbb{E}\big[\text{sim}(X, Z)\big]$.

Fix any leaf $X$, its corresponding node is $c$.

Let $\text{LCA}(X, Y) = f_c^{(k)}$, then $Y \in H_c^{(k)}$, and $d_{\text{LCA}(X,Y)} = \text{depth}(f_c^{(k)}) = d_c - k$.

Let $\text{LCA}(X, Z) = f_c^{(m)}$, then $Z \in H_c^{(m)}$, and $d_{\text{LCA}(X,Z)} = \text{depth}(f_c^{(m)}) = d_c - m$.

From the condition $d_{\text{LCA}(X,Y)} > d_{\text{LCA}(X,Z)}$, we get:

$$d_c - k > d_c - m \implies -k > -m \implies k < m.$$

- Since $k < m$ and $Y \in H_c^{(k)}$, $Z \in H_c^{(m)}$, according to equation (x) (take $i = k$, $j = m$), we have:

$$\mathbb{E}\big[\text{sim}(X, Y)\big] > \mathbb{E}\big[\text{sim}(X, Z)\big].$$

This is exactly the conclusion required by inequalities 11.

In summary, inequalities $10 \Rightarrow$ inequalities 11 holds.

### D.2.2 Part II: inequalities $11 \Rightarrow$ inequalities 10

Assume inequalities 11 holds. It is required to prove that for any leaf class $X$ (corresponding node $c$), and any $0 \le i < j \le d_c$, $Y \in H_c^{(i)}$, $Z \in H_c^{(j)}$, we have $\mathbb{E}\big[\text{sim}(X, Y)\big] > \mathbb{E}\big[\text{sim}(X, Z)\big]$.

Fix any leaf $X$, its corresponding node is $c$.

By the definition of $Y \in H_c^{(i)}$, $\text{LCA}(X, Y) = f_c^{(i)}$, hence $d_{\text{LCA}(X,Y)} = \text{depth}(f_c^{(i)}) = d_c - i$.

By the definition of $Z \in H_c^{(j)}$, $\text{LCA}(X, Z) = f_c^{(j)}$, hence $d_{\text{LCA}(X,Z)} = \text{depth}(f_c^{(j)}) = d_c - j$.

From $i < j$, we obtain:

$$d_c - i > d_c - j \implies d_{\text{LCA}(X,Y)} > d_{\text{LCA}(X,Z)}.$$

According to inequalities 11, we have:

$$\mathbb{E}\big[\text{sim}(X, Y)\big] > \mathbb{E}\big[\text{sim}(X, Z)\big].$$

This is exactly the conclusion required by inequalities 10.

In summary, inequalities $11 \Rightarrow$ inequalities 10 holds.

## E  K-Means Based Database Compression

The inference time and memory consumption of KNN-based classification scale linearly with the size of the retrieval database, making it increasingly costly as the database grows. Moreover, we observe a large number of redundant or highly similar samples within the database, which not only increases

storage and computation costs, but also biases the decision boundary toward sample-dense regions, resulting in distorted classification surfaces.

To address this issue, we introduce a database compression strategy based on K-Means clustering. The goal is to replace original class-specific samples with a small number of representative vectors, thereby reducing computational overhead while smoothing the decision boundary. We assume each representative sample approximates a local spherical region within the original data distribution. Specifically, we apply K-Means clustering to each class independently, partitioning its samples into $k$ clusters.

Consider one such cluster $C_j = \{x_1, x_2, \ldots, x_n\} \subset \mathbb{R}^d$, where all samples $x \in C_j$ are $\ell_2$-normalized embeddings (i.e., $\|x\| = 1$). We aim to find a representative vector $\tilde{x}_j \in \mathbb{R}^d$ that maximizes the total cosine similarity with all samples in the cluster. The objective is formulated as:

$$\max_{\tilde{x}_j \in \mathbb{R}^d} \sum_{x \in C_j} \cos(\tilde{x}_j, x) = \sum_{x \in C_j} \frac{\tilde{x}_j^\top x}{\|\tilde{x}_j\|}. \tag{12}$$

Since we only care about the direction of $\tilde{x}_j$, we constrain it to lie on the unit sphere, i.e., $\|\tilde{x}_j\| = 1$. This simplifies the optimization to:

$$\max_{\|\tilde{x}_j\|=1} \sum_{x \in C_j} \tilde{x}_j^\top x = \tilde{x}_j^\top \left( \sum_{x \in C_j} x \right). \tag{13}$$

The optimum is achieved when $\tilde{x}_j$ is aligned with the direction of the sum vector, yielding:

$$\tilde{x}_j = \frac{\sum_{x \in C_j} x}{\left\| \sum_{x \in C_j} x \right\|}. \tag{14}$$

We thus use the normalized mean vector of each cluster as its representative. As shown in Figure 5, this compression strategy not only reduces classification cost but also improves performance, particularly in scenarios with highly imbalanced class distributions such as RealBench.

## F   More Results on Hybrid Text Detection

Table 5 presents the model's performance on hybrid text detection under OOD settings across different languages in the HART dataset. Despite being trained primarily on English data, with only limited multilingual samples from M4, the model demonstrates strong cross-lingual generalization. This suggests that modeling the generation mechanism helps overcome the limitations of language scarcity.

Figure 8 presents the domain-wise visualization, where pure human-written texts consistently cluster within each domain. In contrast, Figure 2 shows the joint projection across all domains, where human-written texts form four distinct clusters, while AI-generated texts form two clusters. This suggests that text semantics influence generation patterns, with a stronger effect on human writing than on AI generation.

Figure 9 shows the visualization of the HAT structure for fine-grained categories in the essay domain of the HART dataset. Even on the OOD dataset HART, the HAT retains the properties observed in Section 4.2. Moreover, the HAT structure provides an intuitive and interpretable view of the relationships among categories.

## G   Three-Author Text Detection

To better align with realistic scenarios involving more than two authors, we have expanded the HART dataset to include three authors and conducted the following experiments.

We used models not seen during training, such as gemma-3-12b [66] and deepseekv2 [67], along with new prompts to introduce a third author for text editing in the HART dataset. We designed experiments to explore the impact of different editor sequences on the final text. DETree is trained on RealBench and evaluated using the development set as the retrieval database.

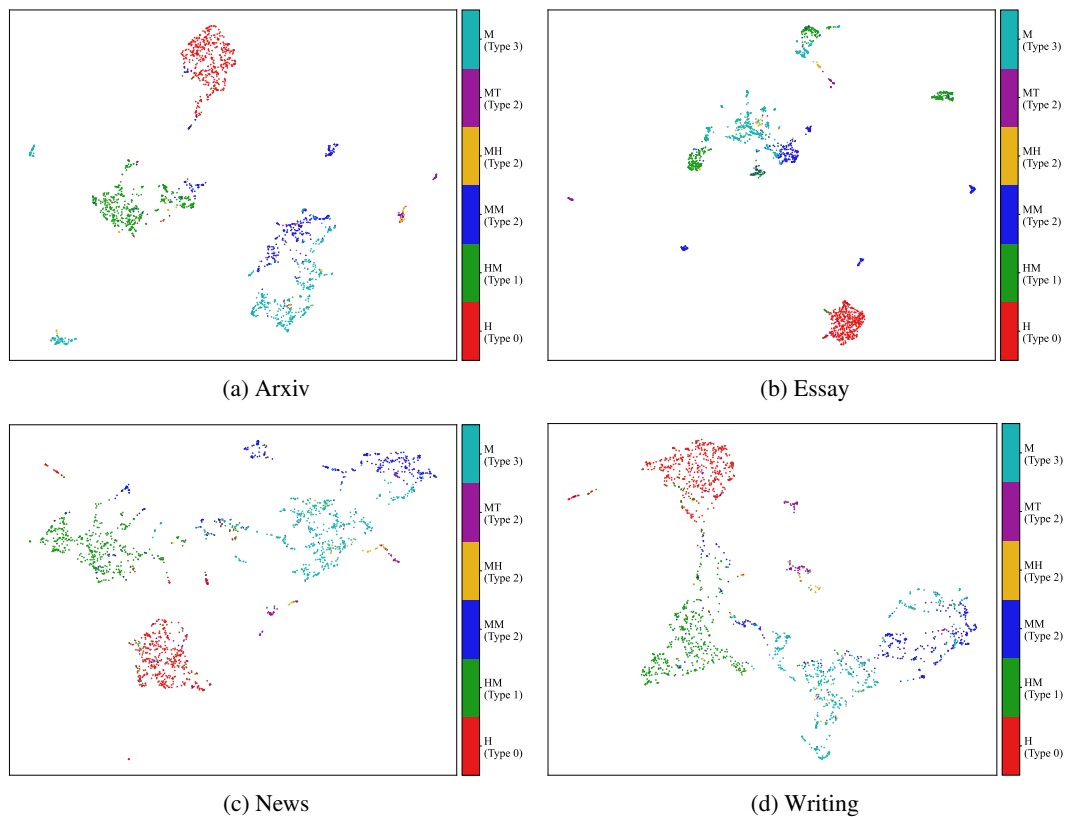

(a) Arxiv

(b) Essay

(c) News

(d) Writing

Figure 8: UMAP-based unsupervised visualization illustrating the distribution of four distinct domains (Arxiv, Essay, News, Writing) from the HART dataset in the representation space of DETree. Colors indicate different text source types, with type indices corresponding to the hierarchical categorization defined in HART. Abbreviations of category names are defined in Figure 3.

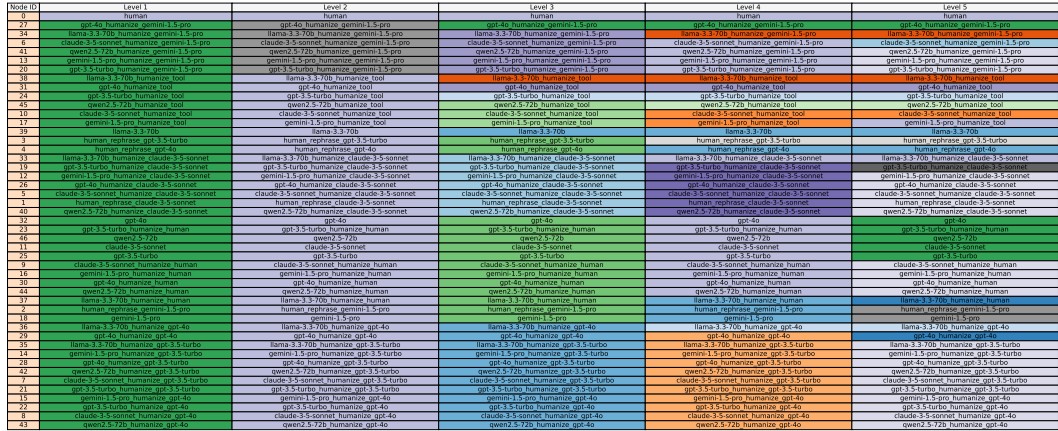

Figure 9: Visualization of the Hierarchical Affinity Tree (HAT) for the detailed categories in the essay domain of the HART dataset. In the figure, Level x represents the corresponding depth level within the HAT. If the current classes have not been split at a particular level, they are colored uniformly. For a class that has already become a leaf node at an earlier level and thus has no corresponding label at a subsequent level in the actual HAT, it inherits both the label and color from its immediate parent node.

Table 5: Results in CCNews of HART, covering five languages. The best AUROC and TPR5% are marked in bold. The column 'ALL' denotes a mixture of languages. DETree is trained on RealBench under prior1 and evaluated using the HART development set as the retrieval database. HART(Fast-Detect / Binoculars / Glimpse) employs the development set to fit its two-dimensional binary classifier.

| Detector | English | Chinese | French | Spanish | Arabic | ALL (TPR5%) |
|---|---|---|---|---|---|---|
| **Level-3** | | | | | | |
| LRR | 0.8466 | 0.8625 | 0.8706 | 0.8744 | 0.6117 | 0.7651 (21%) |
| Fast-Detect | 0.8551 | 0.8655 | 0.8662 | 0.8310 | 0.5871 | 0.8118(48%) |
| Binoculars | 0.8698 | 0.8698 | 0.8814 | 0.8474 | 0.5754 | 0.7990(48%) |
| Glimpse | 0.8310 | **0.8868** | 0.8793 | 0.8382 | 0.7950 | 0.8323(51%) |
| Hart(Fast-Detect) | 0.8600 | 0.8459 | 0.8538 | 0.8397 | 0.5879 | 0.8065 (48%) |
| Hart(Binoculars) | 0.8698 | 0.8495 | 0.8587 | 0.8548 | 0.5476 | 0.7924 (49%) |
| Hart(Glimpse) | 0.8257 | 0.8681 | 0.8853 | 0.8729 | 0.8031 | 0.8481 (53%) |
| **DETree(Ours)** | **0.9888** | 0.8513 | **0.9525** | **0.9468** | **0.9026** | **0.9305 (74%)** |
| **Level-2** | | | | | | |
| LRR | 0.5296 | 0.8748 | 0.7118 | 0.7644 | 0.4777 | 0.6380 (11%) |
| Fast-Detect | 0.6665 | 0.8361 | 0.7728 | 0.6961 | 0.4658 | 0.7007 (37%) |
| Binoculars | 0.6770 | 0.8383 | 0.7779 | 0.7115 | 0.4543 | 0.6929 (37%) |
| Glimpse | 0.5953 | 0.8123 | 0.7511 | 0.7269 | 0.6813 | 0.6921 (33%) |
| Hart(Fast-Detect) | 0.8242 | 0.8295 | 0.8344 | 0.7837 | 0.5867 | 0.7793 (42%) |
| Hart(Binoculars) | 0.8310 | 0.8234 | 0.8464 | 0.7955 | 0.4978 | 0.7515 (38%) |
| Hart(Glimpse) | 0.7094 | 0.8258 | 0.8257 | 0.8083 | 0.7969 | 0.7776 (41%) |
| **DETree(Ours)** | **0.9940** | **0.9538** | **0.9806** | **0.9832** | **0.9374** | **0.9702 (82%)** |
| **Level-1** | | | | | | |
| LRR | 0.5009 | 0.8309 | 0.6480 | 0.7288 | 0.4760 | 0.6070 (08%) |
| Fast-Detect | 0.6897 | 0.8349 | 0.7510 | 0.7331 | 0.4359 | 0.7032 (30%) |
| Binoculars | 0.6969 | 0.8394 | 0.7484 | 0.7461 | 0.4286 | 0.7053 (33%) |
| Glimpse | 0.5600 | 0.7928 | 0.6933 | 0.7034 | 0.6673 | 0.6596 (24%) |
| Hart(Fast-Detect) | 0.7770 | 0.7997 | 0.7749 | 0.7669 | 0.4798 | 0.7288 (32%) |
| Hart(Binoculars) | 0.7843 | 0.8041 | 0.7637 | 0.7657 | 0.4639 | 0.7264 (33%) |
| Hart(Glimpse) | 0.6386 | 0.7904 | 0.7336 | 0.7634 | 0.7638 | 0.7302 (24%) |
| **DETree(Ours)** | **0.9961** | **0.9964** | **0.9945** | **0.9968** | **0.9875** | **0.9936 (99%)** |

First, we investigated the influence of the first author on the final text, while keeping the second and third authors constant. The results in the first author impact section of Table 6 show that when the third author is introduced, the model's ability to distinguish the first author is reduced. However, the model can still effectively differentiate between the two text types.

Second, we also examined the influence of the middle editor on the final text. The results in the second author impact section of Table 6 show a significant decline in detection accuracy when the third author is introduced.

Finally, the third editor is easily distinguishable.

The initial creator of the text sets the general direction and ideas, so these features are difficult to mask through editing. However, the middle editor's features, which are based on rephrasing the initial text, are easily masked by the third editor, leading to a significant decrease in detection accuracy.

This demonstrates that the DETree's encoder is capable of capturing fine-grained, generalizable features. Even when three authors are introduced into the test text (while the model was only exposed to two authors during training), and the third author's model, and prompt are unseen, the encoder can still extract key features to distinguish between different authors.

Table 6: Three-author detection results on the expanded HART dataset using DETree. Each row reports the AUC-ROC of DETree when distinguishing **Source1** from **Source2**. Results are grouped by scenario: (i) *First author impact*—second and third editors are fixed while the first editor varies ; (ii) *Second author impact*—first and third editors are fixed while the middle editor varies ; (iii) *Third author impact*—first and second editors are fixed while the third editor differs.

| Source1 | Source2 | DETree(AUC-ROC) |
|---|---|---|
| **First author impact (GPT-4o vs human)** | | |
| gpt-4o_humanize_gpt-3.5-turbo_ polish_gemma-3 | human_rephrase_gpt-3.5-turbo_ polish_gemma-3 | 96.99 |
| gpt-4o_humanize_gpt-3.5-turbo_ polish_deepseekv2 | human_rephrase_gpt-3.5-turbo_ polish_deepseekv2 | 96.21 |
| gpt-4o_humanize_gpt-3.5-turbo | human_rephrase_gpt-3.5-turbo | 100.00 |
| gpt-4o_polish_gemma-3-12b | human_polish_gemma-3-12b | 99.43 |
| gpt-4o_polish_deepseekv2 | human_polish_deepseekv2 | 99.37 |
| **First author impact (Claude-3.5-sonnet vs human)** | | |
| claude-3-5-sonnet_humanize_ gemini-1.5-pro_polish_gemma-3 | human_rephrase_gemini-1.5-pro_ polish_gemma-3 | 84.17 |
| claude-3-5-sonnet_humanize_ gemini-1.5-pro_polish_deepseekv2 | human_rephrase_gemini-1.5-pro_ polish_deepseekv2 | 84.05 |
| claude-3-5-sonnet_humanize_gemini-1.5-pro | human_rephrase_gemini-1.5-pro | 98.14 |
| claude-3-5-sonnet_polish_gemma-3-12b | human_polish_gemma-3-12b | 98.81 |
| claude-3-5-sonnet_polish_deepseekv2 | human_polish_deepseekv2 | 98.98 |
| **Second author impact (GPT-3.5-turbo vs Claude-3.5-sonnet)** | | |
| gpt-4o_humanize_gpt-3.5-turbo_ polish_gemma-3 | gpt-4o_humanize_claude-3-5-sonnet_ polish_gemma-3-12b | 76.50 |
| gpt-4o_humanize_gpt-3.5-turbo_ polish_deepseekv2 | gpt-4o_humanize_claude-3-5-sonnet_ polish_deepseekv2 | 82.05 |
| gpt-4o_humanize_gpt-3.5-turbo | gpt-4o_humanize_claude-3-5-sonnet | 98.78 |
| **Second author impact (GPT-4o vs Gemini-1.5-Pro)** | | |
| human_rephrase_gpt-4o_ polish_gemma-3 | human_rephrase_gemini-1.5-pro_ polish_gemma-3 | 68.71 |
| human_rephrase_gpt-4o polish_deepseekv2 | human_rephrase_gemini-1.5-pro_ polish_deepseekv2 | 68.02 |
| human_rephrase_gpt-4o | human_rephrase_gemini-1.5-pro | 99.49 |
| **Third author impact (Deepseekv2 vs Gemma-3)** | | |
| qwen2.5-72b_humanize_ claude-3-5-sonnet_polish_gemma-3 | qwen2.5-72b_humanize_ claude-3-5-sonnet_polish_deepseekv2 | 100.00 |
| llama-3.3-70b_humanize_ gemini-1.5-pro_polish_gemma-3 | llama-3.3-70b_humanize_ gemini-1.5-pro_polish_deepseekv2 | 97.69 |
| human_rephrase_gpt-4o_ polish_gemma-3 | human_rephrase_gpt-4o_ polish_deepseekv2 | 99.60 |

# H Benchmark

In this section, we provide a detailed overview of the benchmark datasets used in our experiments.

**MAGE [16]** MAGE is a large-scale benchmark for AI-generated text detection, comprising texts from 27 diverse large language models, including families such as OpenAI GPT [68], LLaMA [62], GLM-130B [63], FLAN-T5 [64], OPT [69], BigScience [70], and EleutherAI [71]. It spans 10 domains and includes 332K training samples and 57K test samples. Additionally, MAGE provides a subset for out-of-domain generalization testing, generated by GPT-4 [12] across four new domains (CNN/DailyMail, DialogSum, PubMedQA, and IMDb), to evaluate model performance in the "Unseen Domains & Unseen Model" scenario. Based on this OOD text, a paraphrasing attack test set was also created using GPT-3.5-turbo [72].

**M4, M4GT [46, 47]** The M4 dataset is a large-scale collection spanning multiple domains, models, and languages, consisting of text generated by 8 large language models across 6 domains and 9 languages. Built upon M4, the M4GT benchmark is designed for AI-generated text detection, with its test set paraphrased using the OUTFOX method to increase task complexity. M4GT defines two task settings: monolingual and multilingual. The monolingual setting contains 120K training and 34K test samples, while the multilingual setting includes 157K training and 42K test samples. In this study, we adopt the M4GT splits for training and evaluation.

**RAID [49]** RAID is a large-scale benchmark dataset for robust machine-generated text detection, comprising over 600K original texts generated by 11 language models across 8 domains. These texts are further expanded to 6.2M samples through 11 types of adversarial attacks. In this study, we use the 605K unperturbed samples released by RAID to construct the training and validation splits for our RealBench dataset.

**TuringBench [48]** TuringBench, released in 2021, is one of the earlier datasets for AI-generated text detection. It focuses on political news headlines and content within a single domain, incorporating texts generated by 19 large language models, including the GPT series [68], GROVER [73], CTRL [74], XLM [75], and XLNet [76]. The dataset consists of 112K training samples and 37K test samples.

**OUTFOX [37]** OUTFOX constructs a student essay detection dataset comprising 15.4K human-written essays and 15.4K LLM-generated essays. It also includes adversarially paraphrased samples generated using DIPPER [77] and OUTFOX attack methods.

**DetectRL [17]** DetectRL is a benchmark designed to evaluate AI-generated text detection in real-world scenarios. It covers four domains with high LLM usage and includes texts generated by GPT-3.5 [72], Claude [78], PaLM-2 [79], and LLaMA-2 [62], along with four types of attacks. We use DetectRL to study the generalization performance of detection models.

**Beemo [18]** Beemo serves as an evaluation benchmark, consisting of 19,683 texts, including edited versions produced by experts, Llama3.1-70B [2], and GPT-4o [12]. It covers five representative task types: open-ended generation, rewriting, summarization, open-domain QA, and closed-domain QA.

**HART [11]** HART is a multi-level test set for hybrid text detection, covering four domains (student essays, arXiv introductions, creative writing, and news) and five languages (English, Chinese, French, Spanish, Arabic). It consists of 32K samples generated by six recent LLMs: GPT-3.5-Turbo [72], GPT-4o [12], Claude 3.5 Sonnet [78], Gemini 1.5 Pro [13], LLaMA 3.3-70B-Instruct [2], and Qwen 2.5-72B-Instruct [3]. HART categorizes samples based on the extent of AI involvement in text generation into four types: human-written (Type 0), human content rephrased by AI (Type 1), AI-generated content humanized through various methods (including LLM paraphrasing, human editing, and commercial tools) (Type 2), and fully AI-generated content (Type 3).

Based on this labeling scheme, HART defines three detection task levels: Level 1 detects AI involvement (Type 0 vs. Type 1/2/3), Level 2 detects whether the core content is AI-generated (Type 0/1 vs. Type 2/3), and Level 3 detects whether the text is fully generated by AI (Type 0/1/2 vs. Type

3). In this study, we strictly adhere to these task definitions to evaluate the model's out-of-domain generalization and performance in hybrid text detection scenarios.

# I    RealBench

RealBench is constructed from the unperturbed samples of MAGE, RAID, TuringBench, and OUT-FOX, with additional augmentation via Hybrid Texts. We further apply various perturbation attacks to enhance training and assess model robustness during evaluation. Section I.1 introduces the construction of Hybrid Texts, and Section I.2 details the perturbation attack strategies.The category composition and statistics of RealBench are summarized in Table 7, and the distribution of sample counts across different augmentation types is provided in Table 8.

Table 7: Composition of the RealBench dataset, where Basic text and Hybrid text represent the number of texts in basic and hybrid categories respectively, Basic categories and Hybrid categories denote the number of basic and hybrid categories. In the test set, the number of categories is presented as "x/y", with x being the corresponding category number and y representing the number of categories that only exist in the test set but not in the validation and training sets.

| Dataset Name | Split | Basic text | Hybrid text | Total text | Basic categories | Hybrid categories | Total categories |
|---|---|---|---|---|---|---|---|
| MAGE | train | 1,433,025 | 4,128,555 | 5,561,580 | 28 | 419 | 447 |
| | valid | 255,164 | 809,730 | 1,064,894 | 28 | 419 | 447 |
| | test | 257,232 | 412,137 | 669,369 | 29/1 | 223/140 | 252 |
| M4 | train | 735,497 | 1,714,893 | 2,450,390 | 6 | 74 | 80 |
| | valid | 25,879 | 71,122 | 97,001 | 4 | 29 | 33 |
| | test | 199,419 | 209,768 | 409,187 | 9/3 | 48/31 | 57 |
| TuringBench | train | 548,756 | 414,293 | 963,049 | 20 | 299 | 319 |
| | valid | 93,036 | 53,087 | 146,123 | 20 | 299 | 319 |
| | test | 182,714 | 169,884 | 352,598 | 20/0 | 139/80 | 159 |
| RAID | train | 1,553,366 | 1,943,413 | 3,496,779 | 12 | 191 | 203 |
| | valid | 173,269 | 218,985 | 392,254 | 12 | 191 | 203 |
| OUTFOX | train | 166,808 | 624,319 | 791,127 | 4 | 56 | 60 |
| | test | 2,956 | 1,956 | 4,912 | 1/2 | 4/2 | 5 |
| Total | all | 5,627,121 | 10,772,142 | 16,399,263 | 66 | 1,138 | 1,204 |

Table 8: Composition of adversarial attack samples in RealBench. Each column represents the amount of text generated by different adversarial attacks (including synonym replacement, perplexity attacks, paraphrase generation, etc.), excluding simple format attacks which are dynamically incorporated during training via on-the-fly augmentation.

| Dataset Name | Split | No Attack | Synonym | Perplexity | Paraphrase | Extend | Polish | Translate | Total |
|---|---|---|---|---|---|---|---|---|---|
| MAGE | train | 319,071 | 261,089 | 852,865 | 156,467 | 1,679,724 | 1,679,724 | 612,640 | 5,561,580 |
| | valid | 56,792 | 46,218 | 152,154 | 19,333 | 340,752 | 340,752 | 108,893 | 1,064,894 |
| | test | 58,381 | 46,359 | 152,492 | 19,075 | 170,457 | 113,638 | 108,967 | 669,369 |
| M4 | train | 292,174 | 113,532 | 329,791 | 50,045 | 718,542 | 718,542 | 227,764 | 2,450,390 |
| | valid | 9,000 | 4,587 | 12,292 | 1,795 | 30,000 | 30,000 | 9,327 | 97,001 |
| | test | 76,650 | 26,500 | 96,269 | 15,888 | 59,976 | 68,544 | 65,360 | 409,187 |
| TuringBench | train | 112,204 | 110,794 | 325,758 | 38,981 | 107,352 | 107,352 | 160,608 | 963,049 |
| | valid | 19,051 | 18,833 | 55,152 | 5,718 | 17,550 | 17,550 | 12,269 | 146,123 |
| | test | 37,357 | 36,944 | 108,413 | 11,178 | 74,714 | 74,714 | 9,278 | 352,598 |
| RAID | train | 544,929 | 511,376 | 372,989 | 856,726 | 215,964 | 215,964 | 778,831 | 3,496,779 |
| | valid | 60,781 | 57,054 | 41,781 | 95,887 | 24,714 | 24,714 | 87,323 | 392,254 |
| OUTFOX | train | 57,600 | 56,483 | 52,725 | - | 259,200 | 259,200 | 105,919 | 791,127 |
| | test | 1,000 | 988 | 968 | - | - | - | 1,956 | 4,912 |
| Total | | 1,644,990 | 1,290,757 | 2,553,649 | 1,271,093 | 3,698,945 | 3,650,694 | 2,289,135 | 16,399,263 |

## I.1 Hybrid Text Construction

The construction of Hybrid Texts involves four main approaches: paraphrasing, translation, continuation, and polishing.

**DIPPER Paraphraser**   Using a fine-tuned T5-11B model (DIPPER [80]) for paraphrasing. The output is labeled as `{orgname}_paraphrase_dipper`, where `orgname` denotes the original source.

**Adversarial Paraphraser**   We adopt a paraphrasing strategy similar to that proposed in OUT-FOX [37] to perform adversarial rewriting of LLM-generated texts. For each input, we first use the encoder model trained by Detective [36] as a retriever to retrieve the top 5 most similar LLM and Human texts from its database. We then construct adversarial prompts using the template shown in Figure 10, and generate the rewritten texts with Qwen2.5-7B-Instruct [3]. The output is labeled as `{orgname}_paraphrase_qwen2.5_7b`.

**Translation Paraphraser**   Texts are first translated into Chinese using Qwen2.5-7B-Instruct and into German using Aya-23-8B [81], then translated back into English. The output is labeled as `{orgname}_translate_{transname}`, where `transname` indicates the translation model used.

**Text Continuation**   We retain the first 128 tokens of each text, constrained to no more than half of the original length, and use them as the prefix for continuation. The prompt templates shown in Figure 10 are used to guide generation. Three templates are used for training and validation, while two additional templates are reserved for testing to enhance generalization. Continuations are generated using LLaMA-3.1-8B [2], Mistral-8B [82], InternLM2-5-7B [83], Olmo2-7B [84], GLM4-9B [85], and Qwen2.5-7B for training and validation, and Gemma-2-9B [86] and DeepSeekV2 [67] for testing. The output is labeled as `{orgname}_extend_{extendname}`, where `extendname` denotes the continuation model used.

**Text Polish**   We apply the prompt templates shown in Figure 10 to polish the texts. Each sample in the training and validation sets is processed with one of 13 randomly selected templates, while two distinct templates are used in the test set for evaluation. The polishing models are identical to those used in the Text Continuation setting. The output is labeled as `{orgname}_polish_{polishname}`, where `polishname` denotes the polish model used.

## I.2 Perturbation Attack

Perturbation attacks are categorized into two types: **Word Attack** and **Format Attack**. For Word Attack, perturbed samples are pre-generated as part of the dataset. In contrast, Format Attack samples are generated on-the-fly during training due to their lower computational overhead.

**Word Attack**   This strategy perturbs words that are potentially influential to the detection model by performing targeted replacements. We implement the following two methods:

- **Synonym Attack**: Synonym substitution is performed using a BERT-based [87] language model. For each selected token, we replace it with a candidate word from the top-ranked synonyms based on semantic similarity.
- **Perplexity Attack**: Inspired by the hypothesis in DetectGPT [27]—that model-generated texts reside near local minima of the language model's loss surface—we generate adversarial samples by perturbing tokens to increase the negative log-likelihood (NLL). Specifically, we use GPT-Neo-2.7B [88] to compute token-wise probabilities, suppress high-probability tokens, and resample from the modified distribution to produce higher-NLL variants.

To constrain semantic drift, we compute the semantic similarity between the original and perturbed texts using a BERT-based model, and discard low-similarity samples.

**Format Attack**   This strategy perturbs textual formats rather than content. We adopt attack schemes primarily derived from the RAID dataset, including:

- **Alternative Spelling**: Replacing American English spellings with their British equivalents.

- **Article Deletion**: Removing articles (e.g., "the", "a", "an").
- **Add Paragraph**: Inserting paragraph delimiters (e.g., "\n") between sentences.
- **Upper-Lower**: Swapping the case of letters within words.
- **Zero-Width Space**: Inserting zero-width spaces (Unicode U+200B) between characters.
- **Whitespace**: Adding whitespace between characters.
- **Homoglyph**: Replacing characters with visually similar Unicode homoglyphs (e.g., replacing "e" with Cyrillic "e" (U+0435)).
- **Number Shuffle**: Randomly shuffling digits in numeric tokens.
- **Misspelling**: Introducing common spelling errors.

| Prompt Type | | Prompt Description |
|---|---|---|
| **Adversarial Paraphraser** | | Here are the results of detecting whether each essay from each problem statement is generated by a Human or a Language Model(LM). 
 Text: {LLM_text1}  Answer: LLM 
 Text: {Human_text1}  Answer: Human 
 ... 
 Your Task: Please rephrase the input LLM-generated text to ensure it is indistinguishable from human-written content. It is essential to maintain the original meaning, tone, and details of the text, while keeping the word count within {input_text_len} words and expressing a clear opinion. 
 Input Text: {input_text} |
| **Translation Paraphraser** | English to Chinese | 请将下列文本从英文翻译成中文，不要有任何额外的输出。英文文本：{input_text} 
 中文文本： |
| | Chinese to English | Please translate the following text from Chinese to English without any additional output. 
 Chinese text: {input_text}  English text: |
| | English to German | Bitte übersetzen Sie den folgenden Text ohne zusätzliche Ausgabe vom Englischen ins Deutsche. 
 Englischer Text: {input_text}  Deutscher Text: |
| | German to English | Please translate the following text from German to English without any additional output. 
 German text: {input_text}  English text: |
| **Text Continuation** | Train&Valid | Here is a piece of text. Please continue writing from where it ends, maintaining the same tone, style, and context while making the continuation coherent and engaging. 
 Input Text: {input_text} |
| | Test | Please continue the following text, expanding on its ideas in a way that maintains a consistent tone and style. The expansion should be coherent, logically structured, and serve to enrich the original content. Avoid using transitional phrases such as 'firstly,' 'secondly,' or 'then.' Instead, opt for smoother transitions that flow naturally from one thought to the next. Use punctuation carefully, particularly minimizing the overuse of commas. 
 Input Text: {input_text} |
| **Text Polish** | Train&Valid | Please refine the following paragraph to improve its flow and clarity. Ensure that the original meaning and structure are preserved, while enhancing sentence construction and expression for better readability: {input_text} |
| | Test | Please adjust the language style of the following paragraph to make it more informal. Maintain the core meaning and structure while ensuring that the tone aligns with a more casual audience. 
 Input Text: {input_text} |

Figure 10: Prompt templates used for Hybrid Text construction, covering four strategies: Adversarial Paraphraser, Translation Paraphraser, Text Continuation, and Text Polish. For Text Continuation and Text Polish, only one representative template is shown.

# J   HAT Visualization

This section presents the intermediate and final results of the HAT construction. After the first stage of supervised contrastive learning, we compute pairwise class similarities using equation 9. Figure 11 visualizes the similarity matrix for all 1,109 classes in the RealBench dataset. While the training objective aims to separate class representations, certain classes remain highly similar due to overlapping textual styles—such as shared sources or common post-editing models. This underscores the importance of modeling inter-class stylistic relationships in AI-generated text detection.

Using the similarity matrix, we apply a hierarchical clustering algorithm [89] to construct an initial dendrogram (binary tree), as shown in Figure 12 under Prior 1. We then incorporate task-specific priors via Algorithm 1 to transform the binary tree into a semantically coherent multiway HAT structure. Figures 13, 14, and 15 illustrate the resulting HAT trees under Priors 1, 2, and 3, respectively. Due to space limitations, we retain all human-related classes and randomly sample 20% of the remaining classes for HAT construction and visualization.

In Figures 13, 14, and 15, each Level represents a depth layer in the HAT tree. At each level, all categories belonging to the same node are shown in the same color. For nodes that became leaf nodes at earlier levels and are not further split, we replicate their information in subsequent levels and retain their original color for consistency. For illustration purposes, only the top 5 levels are shown.

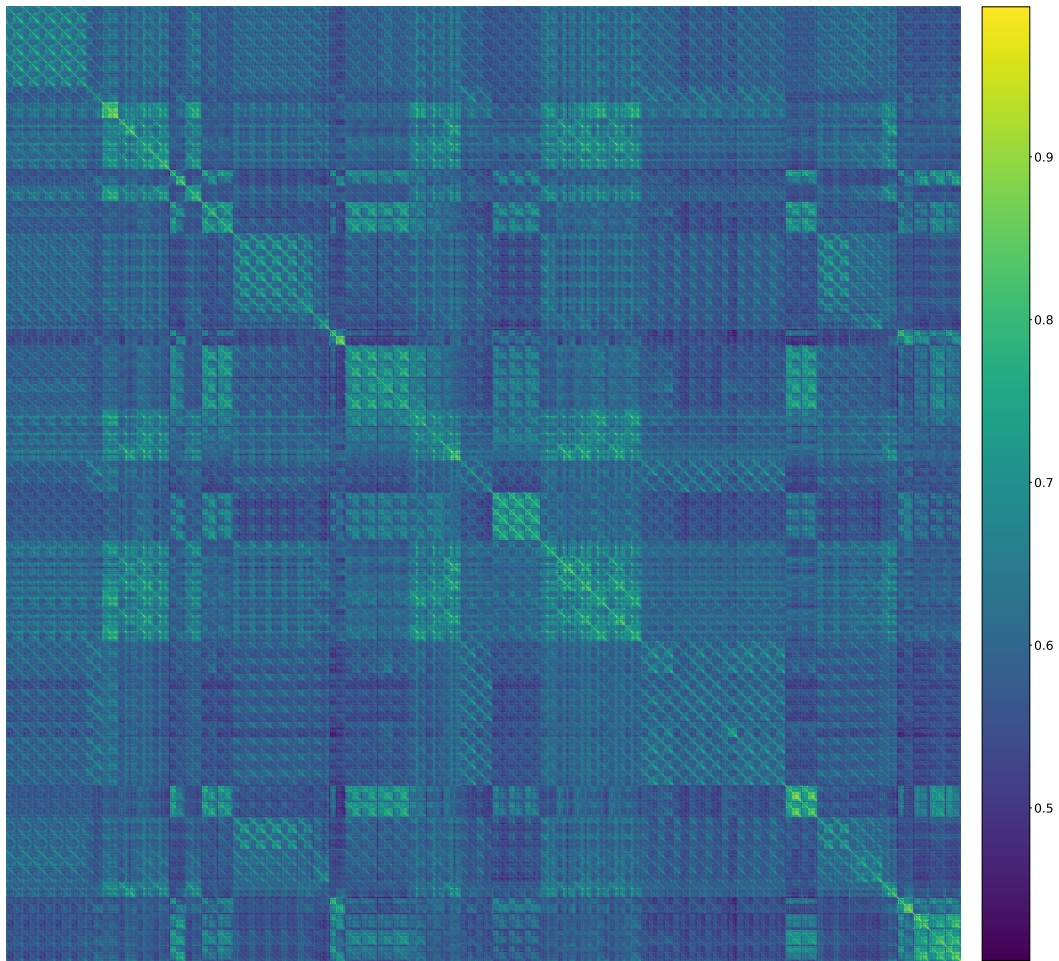

Figure 11: Similarity matrix of 1,109 classes in the RealBench training set after Stage 1 supervised contrastive learning. Brighter regions indicate higher similarity.

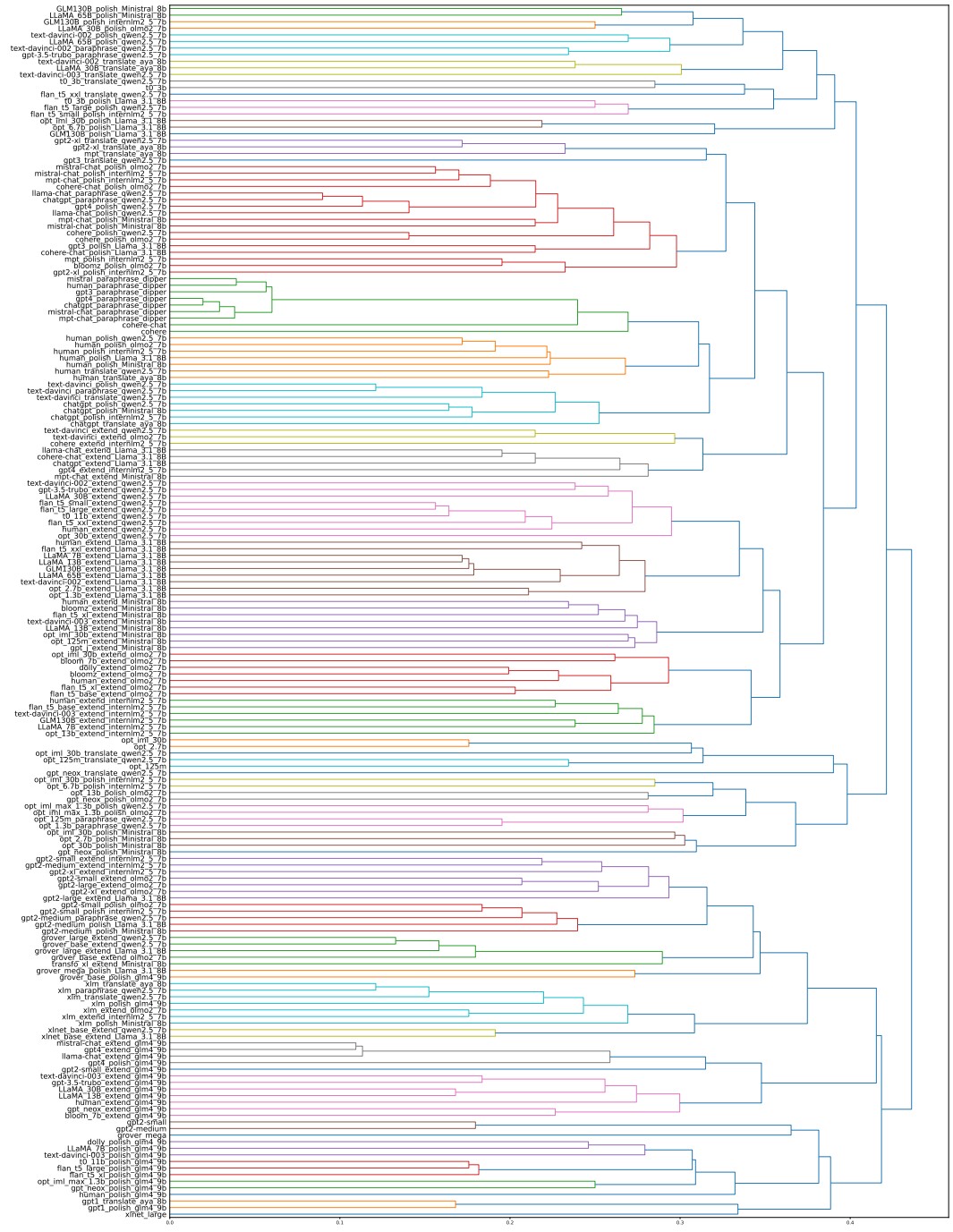

Figure 12: Dendrogram constructed from the similarity matrix under Prior 1, based on a randomly selected 20% subset of categories from the RealBench training set.

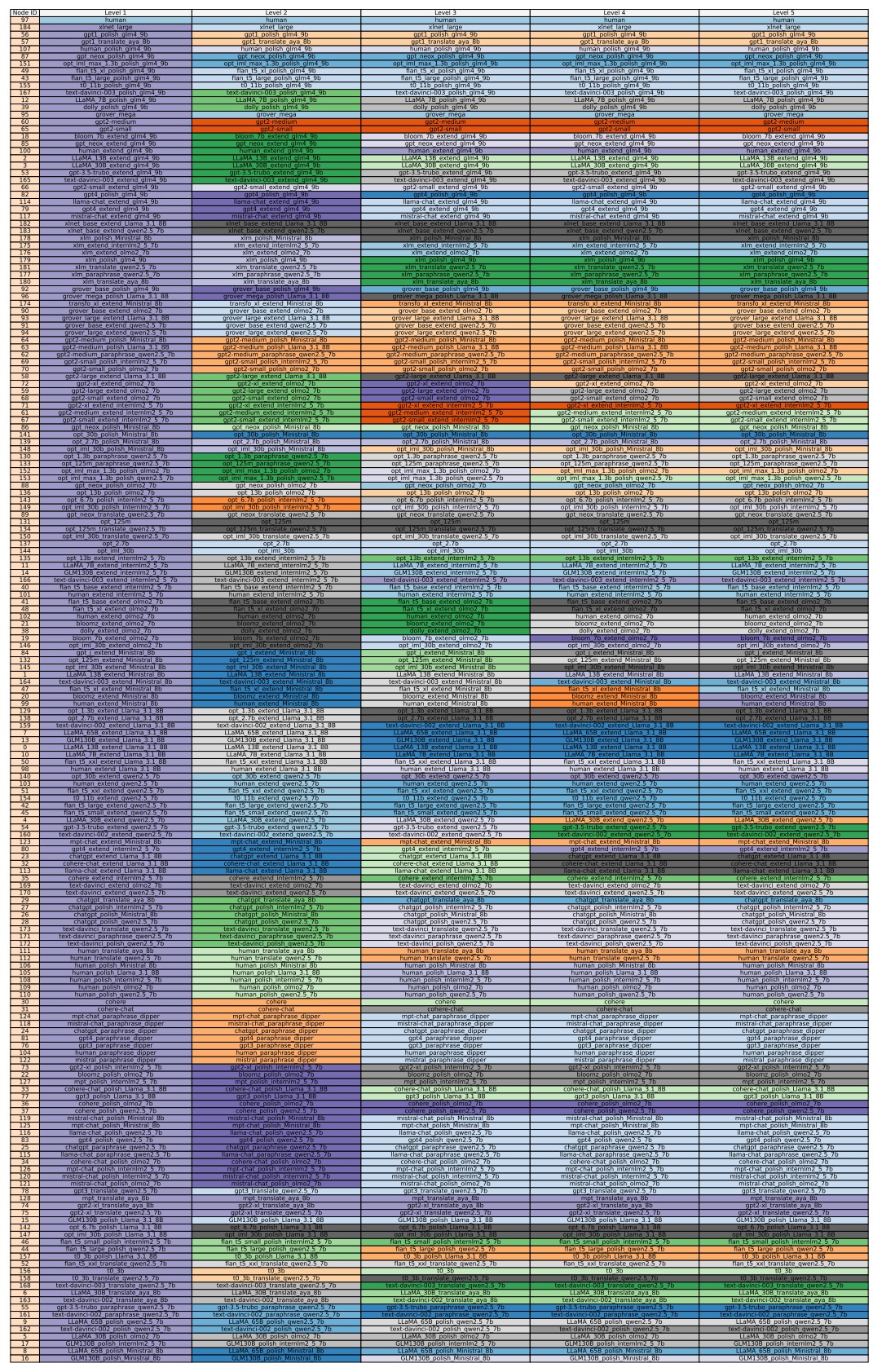

Figure 13: Visualization of the HAT constructed under Prior1 using 20% randomly sampled Real-Bench training categories.

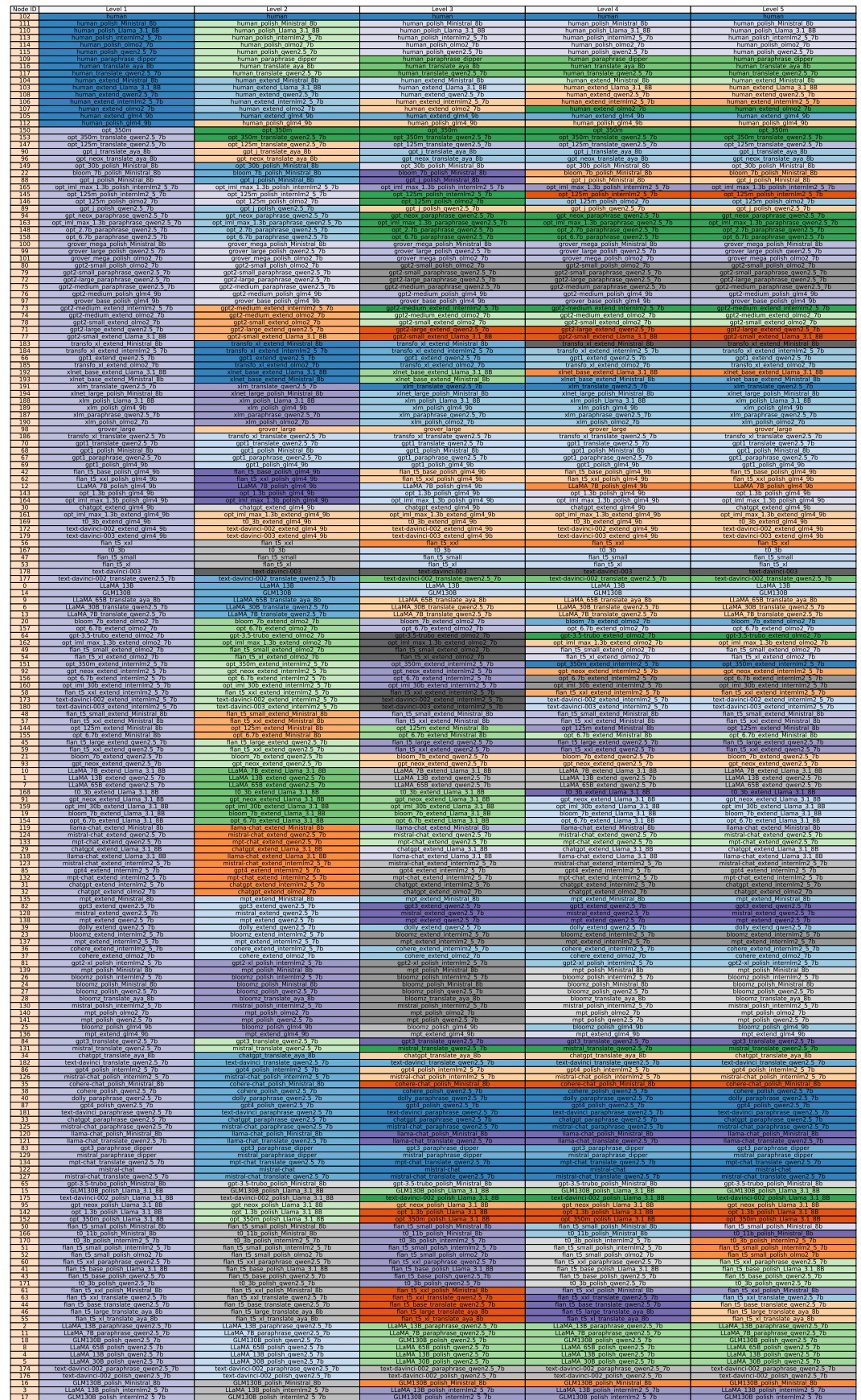

Figure 14: Visualization of the HAT constructed under Prior2 using 20% randomly sampled Real-Bench training categories.

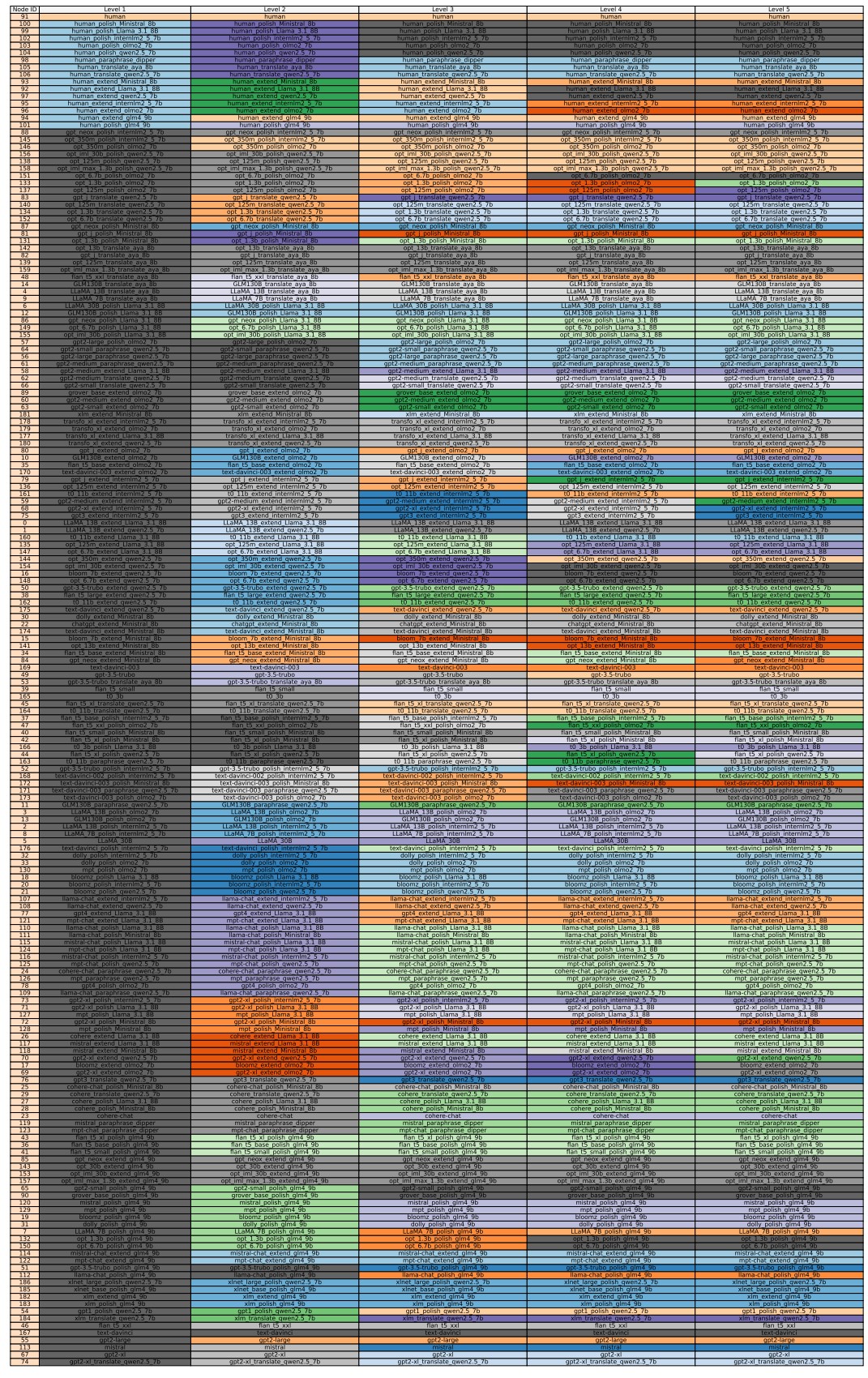

Figure 15: Visualization of the HAT constructed under Prior3 using 20% randomly sampled Real-Bench training categories.

