# OpenReview forum: "DETree: DEtecting Human-AI Collaborative Texts via Tree-Structured Hierarchical Representation Learning"
_NeurIPS.cc/2025/Conference — NeurIPS 2025 poster_

### Official Review · Reviewer_DGKn · 2025-07-02

**Clarity:** 3
**Significance:** 4
**Originality:** 3
**Rating:** 5
**Confidence:** 3

**Summary:**

The work "DETree: DEtecting Human-AI Collaborative Texts via Tree-Structured Hierarchical Representation Learning" introduces DETree, a detector for hybrid (human-AI collaborative) text based on representation learning. The paper tackles an already very important problem of identifying how large language models (LLMs) participate in a text's creation rather than merely deciding "human vs. machine" as a classification task. The key contributions are:
* RealBench, a new benchmark with 16.4M samples spanning 1204 fine‑grained categories that encode generation pipelines (e.g., Llama‑3 draft polished by GPT‑4o);
* a new approach to identifying these categories as follows:
    * construct a Hierarchical Affinity Tree (HAT) over 1200+ source categories that arise in RealBench;
    * train the encoder with a Tree‑Structured Contrastive Loss (TSCL) that forces embeddings to respect that hierarchy,
    * then perform k‑NN classification with a compressed, centroids‑only database.

Experiments cover in‑distribution, OOD, adversarial and few‑shot settings. DETree convincingly beats a strong set of baselines (RoBERTa, T5‑Sentinel, DeTeCTive, Fast‑DetectGPT, Binoculars, HART, UAR) on four public benchmarks and in several new scenarios, showing strong empirical gains.

**Questions:**

1. How many RealBench samples are verbatim or near‑duplicate across training/validation/test? Please provide a de‑duplication analysis.
2. Did you test DETree with an entirely unseen LLM that does not appear in RealBench?
3. What is the exact setting of the Silhouette‑based subtree edit heuristic (threshold, max depth)? An ablation varying these parameters would clarify its importance.
4. Could TSCL be combined with a lightweight classifier (e.g., logistic regression on frozen embeddings) to remove the dependency on k‑NN?

**Ethical Concerns:**

["NO or VERY MINOR ethics concerns only"]

**Final Justification:**

I really liked the detailed author responses (to me and to other reviewers) but my score is already at "Accept", and it's not *that* groundbreaking a paper, so I'll leave it at that.

**Limitations:**

The authors mention the scarcity of >2‑author collaborations and rare domains , but should also discuss:
* the synthetic nature and potential style bias of RealBench;
* relatively heavy GPU and memory footprintl;
* how easy it would be for sophisticated attackers to fine‑tune their style to evade tree‑aware detectors (in the context of the detection-generation arms race).

**Paper Formatting Concerns:**

The text on figures 3, 10, 12-15 is tiny to the point of unreadability (especially in the Appendix). Consider breaking them up into more figures to increase the font size. Minor issues with text consistency, e.g,. capiltalized "in Equation 3" on l.174 but immediately "in equation 4" on l.175.

**Quality:**

3

**Strengths And Weaknesses:**

Strengths:
* [*significance*] a large carefully engineered new dataset, which is likely to become a new and important benchmark;
* [*originality*] as far as I know, this is the first paper to explicitly model hierarchical provenance for hybrid text detection; this is also a good point for significance as hybrid provenance detection is of high practical and societal relevance;
* [*originality*] TSCL is a non‑trivial extension of supervised contrastive learning;
* [*quality*] the results are complemented by theoretical proofs;
* [*significance*] the experimental study is convincing, with a detailed ablation study;
* [*clarity*] the writing is generally clear and well organized; the figures (UMAP, HAT visualizations) are intuitive and helpful.

Weaknesses:
* [*quality*] RealBench is largely synthetic (paraphrase/translation of existing corpora); the authors do not quantify annotation noise or genre imbalance;
* [*originality*] the approach uses well‑known building blocks (contrastive learning, agglomerative clustering, k‑NN), so novelty is incremental;
* [*significance*] RealBench is also used as the retrieval database at test time, so in‑distribution and some OOD scores may be optimistic.

---

> ### Author Rebuttal · Authors · 2025-07-31
>
> *We sincerely thank the reviewer DGKn for the thoughtful and detailed comments. Below, we provide point-by-point responses to each of the concerns.*
> ***
> >Weakness1: [*quality*] RealBench is largely synthetic (paraphrase/translation of existing corpora); the authors do not quantify annotation noise or genre imbalance;
>
> Thank you for your valuable feedback regarding annotation noise and genre imbalance. Our annotation process is based on publicly available datasets and is programmatically generated, ensuring determinism and the absence of additional subjective bias. Therefore, it theoretically does not introduce new annotation noise. In the final version, we will include a detailed explanation of the annotation quality in the original datasets and further examine whether any noise analysis has already been conducted. Additionally, we will provide a breakdown of the genre distribution in RealBench in the appendix.
>
> >Weakness2: [*originality*] the approach uses well‑known building blocks (contrastive learning, agglomerative clustering, k‑NN), so novelty is incremental;
>
> We appreciate the reviewer's comments on the originality of our approach. Our method models the hierarchical relationships between different sources in human-AI collaborative texts using the Hierarchical Affinity Tree(HAT) structure. Building on this, we introduce Tree-structured Contrastive Loss (TSCL), which innovatively applies contrastive learning to the modeling of HAT structures, aligning the embedding space with the HAT structure. This method provides a general and interpretable solution for handling scalable author categories and can be conveniently applied to other hierarchical tasks. Additionally, we propose a K-Means-based compression algorithm and a retrieval-based few-shot method, which significantly enhance both classification speed and accuracy.
> Our approach demonstrates state-of-the-art performance in AI text detection across in-domain, out-of-domain, and hybrid text scenarios. It offers an effective detection solution in the context of the rapid emergence of new LLMs.
>
> >Weakness3: [*significance*] RealBench is also used as the retrieval database at test time, so in‑distribution and some OOD scores may be optimistic.
>
> We would like to provide some additional clarifications regarding the experimental setup:
>
> First, in Table 1, we present multiple sets of supervised training results, including training individually on each dataset and training with different priors on RealBench.
>
> Second, although DTree in Table 2 uses RealBench as the training set, it is tested on multiple retrieval databases, including RealBench, MAGE, M4, and RAID. The results show that even without using RealBench as the retrieval database, the models perform well in OOD environments. Additionally, we conducted few-shot experiments, reporting performance on the same support set for models that support few-shot learning, and providing results without few-shot support for other models to ensure fair comparisons.
>
> Finally, in Table 3, we trained the model solely on RealBench and evaluated it according to the original test setup from the HART dataset, where the retrieval database was not the RealBench training set but the dev split of HART.
>
> > Q1: How many RealBench samples are verbatim or near‑duplicate across training/validation/test? Please provide a de‑duplication analysis.
>
> Thank you for your reminder. We performed a duplication analysis on the texts in RealBench using a hash algorithm. There are no duplicate texts in the original datasets- MAGE, M4, RAID, OUTFOX, or in their corresponding augmented versions. However, in the TuringBench dataset, 663 texts are identical between the training and test sets, and 311 texts are identical between the training and validation sets. A small number of duplicates were also found in the augmented data derived from TuringBench.
>
> > Q2: Did you test DETree with an entirely unseen LLM that does not appear in RealBench?
>
> LLaMA 3.1‑70B and GPT‑4o in Beemo, Claude and PaLM‑2 in DetectRL, GPT‑4o, Claude 3.5 Sonnet, Gemini 1.5 Pro, LLaMA 3.3‑70B‑Instruct, and Qwen 2.5‑72B‑Instruct in HART are all models not seen in RealBench.
>
> > Q3: What is the exact setting of the Silhouette‑based subtree edit heuristic (threshold, max depth)? An ablation varying these parameters would clarify its importance.
>
> Thank you for your insightful comment. In the construction of HAT subtrees, we control the degree of node splitting using a termination threshold. A smaller threshold results in deeper trees. To assess the impact of tree depth on model performance, we tested three different termination thresholds (0.3, 0.25, and 0.2), corresponding to maximum tree depths of 3, 6, and 9, respectively. The model was trained on RealBench and evaluated on MAGE.
>
> The experimental results show that DETree performs well across different tree depths. Specifically, when the tree depth is shallow, fine-grained category information is lost, leading to a performance drop. As the tree depth increases, model performance improves, reaching an optimal level at a depth of 6. Further increasing the depth only results in a slight decrease in performance. The results reported in the paper correspond to the experiment with a maximum tree depth of 6.
>
> |max dep|3|6|9|
> |-|-|-|-|
> |F1|96.48|96.96|96.84|
> |AvgRecall|96.38|96.87|96.76|
>
> We will include additional ablation studies in the final version to further clarify this aspect.
>
> > Q4: Could TSCL be combined with a lightweight classifier (e.g., logistic regression on frozen embeddings) to remove the dependency on k‑NN?
>
> With TSCL training, the fine-grained features learned by DETree are capable of distinguishing between various text categories. As a result, any classifier can be used for classification, and this can be combined with our K-Means Based Database Compression method to further reduce training overhead.
>
> While we chose to use k-NN for classification due to its ability to fit complex decision boundaries and to avoid introducing additional training overhead for the few-shot method.
>
> The following experimental results report that DETree was trained on the MAGE training set, with the K-Means compression method used to reduce the training set to 10K samples for use with traditional classifiers, and tested on the MAGE test set.
>
> ||F1|AvgRecall|
> |-|-|-|
> |KNN|96.96|96.87|
> |SVM|96.63|96.57|
> |Logistic Regression|96.60|96.53|
> |Random Forest|96.60|96.49|
>
> > Limitations:
> >
> > The authors mention the scarcity of >2‑author collaborations and rare domains, but should also discuss: 1.the synthetic nature and potential style bias of RealBench; 2. relatively heavy GPU and memory footprintl; 3. how easy it would be for sophisticated attackers to fine‑tune their style to evade tree‑aware detectors (in the context of the detection-generation arms race).
>
> As for the synthetic nature and potential style bias of RealBench, we employed different paraphrasing models and prompts during both the training and testing phases of RealBench. Additionally, we introduced multiple external datasets with significant differences in style and generation methods (such as MAGE, M4, DetectRL, Beemo, HART, etc.) during evaluation to ensure diversity and generalization.
>
> During the training phase, we use a RoBERTa-large encoding model with approximately 0.5B parameters. Given the large dataset size, training can be completed in about 40 hours using 8×RTX4090 GPUs.
>
> In the inference phase, when we compress the database to 10k samples for k-NN retrieval classification, the computational overhead is equivalent to attaching a 10k-sized classification head (approximately 10M parameters) after the encoder, with minimal memory cost.
>
> Regarding the concern about sophisticated attackers fine-tuning their style to evade tree-aware detectors, this adversarial issue has not been addressed in the current work. We acknowledge the importance of this challenge and will discuss it in the limitations and future work sections of the final version.
>
> > Paper Formatting Concerns:
> >
> > The text on figures 3, 10, 12-15 is tiny to the point of unreadability (especially in the Appendix). Consider breaking them up into more figures to increase the font size. Minor issues with text consistency, e.g,. capiltalized "in Equation 3" on l.174 but immediately "in equation 4" on l.175.
>
> Thank you for the valuable comments on formatting. We will address the above issues in the final version.

---

> > ### Author Response · Authors · 2025-08-06
> >
> > Dear Reviewer DGKn,
> >
> > I hope this message finds you well. As the discussion period is nearing its end, I wanted to ensure we have addressed all your concerns satisfactorily. If there are any additional points or feedback you'd like us to consider, please let us know. Your insights are invaluable to us, and we remain eager to resolve any remaining issues to further improve our work.
> >
> > Thank you again for your time and effort in reviewing our paper.
> >
> > Best wishes,
> >
> > The Authors

---

> > ### Comment · Reviewer_DGKn · 2025-08-08
> >
> > Thank you for your replies! They answer all my questions.

---

### Official Review · Reviewer_U1pr · 2025-07-02

**Clarity:** 3
**Significance:** 3
**Originality:** 3
**Rating:** 4
**Confidence:** 2

**Summary:**

This paper introduces DETree, a method for detecting AI-generated text that is both interpretable and effective. The authors propose to build an interpretable decision tree classifier trained on features derived from both token-level perplexity signals and semantic embedding similarities.

**Questions:**

how does DETree behave when model-generated text has been post-edited by humans or blends human and AI content (i.e., in mixed authorship settings)?

How much does each category of features—local perplexity, global perplexity, embedding similarity—contribute to the model’s performance?

could you compare DETree to other lightweight or interpretable models such as logistic regression, rule-based classifiers, or shallow decision forests?

How do you balance the depth and complexity of the decision tree against interpretability? Does a deeper tree improve performance substantially, or introduce interpretability costs?

**Ethical Concerns:**

["NO or VERY MINOR ethics concerns only"]

**Limitations:**

yes

**Quality:**

3

**Strengths And Weaknesses:**

Strength:
(1) The authors carefully design a suite of features that capture both local (e.g., token-level perplexity) and global (e.g., embedding similarity) signals.
(2) DETree is evaluated across multiple settings—cross-domain, cross-LLM, and adversarial attacks (paraphrasing)—and consistently performs competitively with black-box detectors.
(3) The use of a decision tree as the classifier provides transparent, step-by-step reasoning behind predictions. The authors also provide visualizations of decision paths and feature importance to illustrate interpretability in action.

Weakness:

Certain feature definitions, especially around how local and global perplexity are computed and normalized, could be further clarified clearer mathematical notation or concrete examples.

---

> ### Author Rebuttal · Authors · 2025-07-31
>
> *We sincerely thank the reviewer U1pr for the thoughtful and detailed comments. Below, we provide point-by-point responses to each of the concerns.*
> ***
> > Weakness:Certain feature definitions, especially around how local and global perplexity are computed and normalized, could be further clarified clearer mathematical notation or concrete examples.
>
> Thank you for pointing out the lack of clarity in the expression. You mentioned that "local and global perplexity are computed and normalized, and could be further clarified with clearer mathematical notation or concrete examples." However, we did not use the terms "local and global perplexity" in the paper. To avoid any misunderstandings, we would appreciate further clarification on which specific part you are referring to, so we can make the necessary improvements and additions.
>
> > Q1: how does DETree behave when model-generated text has been post-edited by humans or blends human and AI content (i.e., in mixed authorship settings)?
>
> In mixed authorship settings, we investigate the behavior of DETree in the Hybrid Text Detection section of the paper, based on the HART dataset. HART categorizes samples based on the extent of AI involvement in text  generation into four types: human-written (Type 0), human content rephrased by AI (Type 1), AI generated content humanized through various methods (including LLM paraphrasing, human editing,  and commercial tools) (Type 2), and fully AI-generated content (Type 3).  Based on this labeling scheme, HART defines three detection task levels: Level 1 detects AI  involvement (Type 0 vs. Type 1/2/3), Level 2 detects whether the core content is AI-generated (Type  0/1 vs. Type 2/3), and Level 3 detects whether the text is fully generated by AI (Type 0/1/2 vs. Type 3).
>
> DETree is trained on RealBench and evaluated strictly according to the testing setup of the HART dataset. As shown in Table 3 of the paper, the model performs significantly better than the results reported in the original paper under this setup.
>
> To better align with realistic scenarios involving more than two authors, we have expanded the HART dataset to include three authors and conducted the following experiments:
>
> We used models not seen during training, such as gemma-3-12b and deepseekv2, along with new prompts to introduce a third author for text editing in the HART dataset. We designed experiments to explore the impact of different editor sequences on the final text.
>
> DETree is trained on RealBench and evaluated using the development set as the retrieval database. First, we investigated the influence of the first author on the final text, while keeping the second and third authors constant. The results show that when the third author is introduced, the model's ability to distinguish the first author is reduced. However, the model can still effectively differentiate between the two text types.
>
> |source1|source2|DETree(aucroc)|
> |-|-|-|
> |gpt-4o_humanize_gpt-3.5-turbo_polish_gemma-3|human_rephrase_gpt-3.5-turbo_polish_gemma-3|96.99|
> |gpt-4o_humanize_gpt-3.5-turbo_polish_deepseekv2|human_rephrase_gpt-3.5-turbo_polish_deepseekv2|96.21|
> |gpt-4o_humanize_gpt-3.5-turbo|human_rephrase_gpt-3.5-turbo|100|
> |gpt-4o_polish_gemma-3-12b|human_polish_gemma-3-12b|99.43|
> |gpt-4o_polish_deepseekv2|human_polish_deepseekv2|99.37|
> ||||
> |claude-3-5-sonnet_humanize_gemini-1.5-pro_polish_gemma-3|human_rephrase_gemini-1.5-pro_polish_gemma-3|84.17|
> |claude-3-5-sonnet_humanize_gemini-1.5-pro_polish_deepseekv2|human_rephrase_gemini-1.5-pro_polish_deepseekv2|84.05|
> |claude-3-5-sonnet_humanize_gemini-1.5-pro|human_rephrase_gemini-1.5-pro|98.14|
> |claude-3-5-sonnet_polish_gemma-3-12b|human_polish_gemma-3-12b|98.81|
> |claude-3-5-sonnet_polish_deepseekv2|human_polish_deepseekv2|98.98|
> We also examined the influence of the middle editor on the final text. The results show a significant decline in detection accuracy when the third author is introduced. The model finds it much harder to distinguish between the two text types when the middle editor is involved.
>
> |source1|source2|DETree(aucroc)|
> |-|-|-|
> |gpt-4o_humanize_gpt-3.5-turbo_polish_gemma-3|gpt-4o_humanize_claude-3-5-sonnet_polish_gemma-3-12b|76.50|
> |gpt-4o_humanize_gpt-3.5-turbo_polish_deepseekv2|gpt-4o_humanize_claude-3-5-sonnet_polish_deepseekv2|82.05|
> |gpt-4o_humanize_gpt-3.5-turbo|gpt-4o_humanize_claude-3-5-sonnet|98.78|
> ||||
> |human_rephrase_gpt-4o_polish_gemma-3|human_rephrase_gemini-1.5-pro_polish_gemma-3|68.71|
> |human_rephrase_gpt-4o_polish_deepseekv2|human_rephrase_gemini-1.5-pro_polish_deepseekv2|68.02|
> |human_rephrase_gpt-4o|human_rephrase_gemini-1.5-pro|99.49|
>
> Finally, the third editor is easily distinguishable.
>
> |source1|source2|DETree(aucroc)|
> |-|-|-|
> |qwen2.5-72b_humanize_claude-3-5-sonnet_polish_gemma-3|qwen2.5-72b_humanize_claude-3-5-sonnet_polish_deepseekv2|100|
> |llama-3.3-70b_humanize_gemini-1.5-pro_polish_gemma-3|llama-3.3-70b_humanize_gemini-1.5-pro_polish_deepseekv2|97.69|
> |human_rephrase_gpt-4o_polish_gemma-3|human_rephrase_gpt-4o_polish_deepseekv2|99.6|
>
> The initial creator of the text sets the general direction and ideas, so these features are difficult to mask through editing. However, the middle editor's features, which are based on rephrasing the initial text, are easily masked by the third editor, leading to a significant decrease in detection accuracy.
>
> This demonstrates that the DETree's encoder is capable of capturing fine-grained, generalizable features. Even when three authors are introduced into the test text (while the model was only exposed to two authors during training), and the third author's model, and prompt are unseen, the encoder can still extract key features to distinguish between different authors.
>
> > Q2: How much does each category of features—local perplexity, global perplexity, embedding similarity—contribute to the model’s performance?
>
> In our approach, embedding similarity plays a central role in computing the class similarity matrix, constructing the HAT structure, and performing the final retrieval classification. As for local perplexity and global perplexity, these features were not used in the paper.
>
> > Q3: could you compare DETree to other lightweight or interpretable models such as logistic regression, rule-based classifiers, or shallow decision forests?
>
> In our experiments, we compared DETree with rule-based classifiers designed using perplexity and its variants, such as LRR, Glimpse, Fast-Detect, and Binoculars. These classifiers are lightweight and interpretable. As shown in Tables 1, 2, and 3, DETree outperforms these models in both in-domain, out-of-domain, and hybrid text test environments.
>
> > Q4: How do you balance the depth and complexity of the decision tree against interpretability? Does a deeper tree improve performance substantially, or introduce interpretability costs?
>
> Thank you for your insightful comment. In the construction of HAT subtrees, we control the degree of node splitting using a termination threshold. A smaller threshold results in deeper trees. To assess the impact of tree depth on model performance, we tested three different termination thresholds (0.3, 0.25, and 0.2), corresponding to maximum tree depths of 3, 6, and 9, respectively. The model was trained on RealBench and evaluated on MAGE.
>
> The experimental results show that DETree performs well across different tree depths. Specifically, when the tree depth is shallow, fine-grained category information is lost, leading to a performance drop. As the tree depth increases, model performance improves, reaching an optimal level at a depth of 6. Further increasing the depth only results in a slight decrease in performance. The results reported in the paper correspond to the experiment with a maximum tree depth of 6.
>
> |max dep|3|6|9|
> |-|-|-|-|
> |F1|96.48|96.96|96.84|
> |AvgRecall|96.38|96.87|96.76|
>
> We will include additional ablation studies in the final version to further clarify this aspect.

---

> > ### Author Response · Authors · 2025-08-06
> >
> > Dear Reviewer U1pr,
> >
> > I hope this message finds you well. As the discussion period is nearing its end, I wanted to ensure we have addressed all your concerns satisfactorily. If there are any additional points or feedback you'd like us to consider, please let us know. Your insights are invaluable to us, and we remain eager to resolve any remaining issues to further improve our work.
> >
> > Thank you again for your time and effort in reviewing our paper.
> >
> > Best wishes,
> >
> > The Authors

---

### Official Review · Reviewer_1CCF · 2025-07-03

**Clarity:** 3
**Significance:** 2
**Originality:** 3
**Rating:** 5
**Confidence:** 3

**Summary:**

The paper proposes hierarchical contrastive training to detect collaborative text between humans and AI. The idea is that there are different degrees of similarity between human text, AI-generated text, modified human text using AI, and modified AI-generated text using AI, etc, so the effectively discriminate all cases, it is natural to arrange those classes hierarchically and use this hierarchy when training an embedding models with a contrastive loss.

**Questions:**

- Please fix my comments regarding Theorem 3.1. It is possible that this theorem does not contribute much, and perhaps better to remove it.
- Please explain if you allow few-shot adaptation when evaluating competing methods. This is crucial to understanding whether the improvement is the result of the proposed hierarchical contrastive methodology.

**Ethical Concerns:**

["NO or VERY MINOR ethics concerns only"]

**Final Justification:**

The authors responded satisfactorily to my inquiries. I believe that the changes will improve the paper's clarity.

I increased the paper's rating.

**Limitations:**

Yes

**Quality:**

3

**Strengths And Weaknesses:**

Strengths:
- The representation approach for authorship identification using contrastive learning has drawn significant attention recently. The current paper proposes an interesting extension (hierarchical contrastive).
- The evaluations appear comprehensive as far as I can tell. They demonstrate well that the proposed method outperformed other methods in identifying collaboration cases.

Weaknesses:
- While it appears from the results that we can detect collaboration on datasets different from those in training, the process of creating these simulated test cases is so similar to that in the training that it is difficult to argue that the approach can be applied in new scenarios. This problem is not necessarily technical in the sense that test cases are difficult to gather; it is more related to an attempt to tackle too broad a scenario. To clarify, the data science aspect of the work is solid; this weakness is related to the motivation for detecting human-AI collaboration. Perhaps you can consider different situations where hierarchy is natural, so the methodology can be useful?
- Theorem 3.1 is not stated as a theorem. The sentence "categories with closer LCA are expected to be more similar in the embedding space" is problematic because it is unclear with respect to what distribution the expectation is evaluated. The word "should" in the second line is confusing. Can you remove it?
In Line 247 you refer to the "constraint defined in Theorem 3.1". What constraint? Do you mean "condition"?
- In Retrieval-Based Few-Shots adaptation, it is unclear what you mean by "superficial features" and why their "ability to generalize across domains" is limited. Of course, the performance bottleneck due to differences between train and test cases is the fundamental challenge in learning. It is unclear how the approach taken here is different than standard training. Also, do you allow few-shot adaptation in competing methods?
- Related works: It is worth noting that the problem of detecting collaboration between authors existed before the AI era, and some methods were developed there (for example, Savoy, Jacques. "Machine learning methods for stylometry." Cham: Springer (2020) and related references). Additionally, related to AI text, [Kashtan & Kipnis, Harvard Data Science Review. 2024] proposed a method to address "sparse human-AI collaboration": a few human edits of AI text or vice versa. In general, these references discuss methods that are more explainable as they do not rely on representation learning, as in the current paper.

---

> ### Author Rebuttal · Authors · 2025-07-31
>
> *We sincerely thank the reviewer 1CCF for the thoughtful and detailed comments. Below, we provide point-by-point responses to each of the concerns.*
> ***
> > Weakness1: While it appears from the resutls that we can detect collaboration on datasets different from those in training, the process of creating these simulated test cases is so similar to that in the training that it is difficult to argue that the approach can be applied in new scenrios [...]
>
> Thank you for your positive feedback. In our generalization analysis, we did not solely rely on our own dataset, RealBench, but also incorporated challenging datasets like DetectRL, Beemo (Table 2), and HART (Table 3), which are recently published. Notably, HART contains data generated by real human-AI collaboration. The three levels of testing proposed by HART are particularly relevant to real-world complex application scenarios.
>
> The motivation for our "human-AI collaboration detection" task inherently encompasses a wide range of scenarios and is highly challenging. While existing research and published datasets may not yet cover all real-world collaboration types, the methodology we propose is a general detection method with multi-class hierarchical modeling ability and task adaptability. We believe our approach has the potential to be extended to more complex scenarios.
>
> To better align with realistic scenarios involving more than two authors, we have expanded the HART dataset to include three authors and conducted the following experiments:
>
> We used models not seen during training, such as gemma-3-12b and deepseekv2, along with new prompts to introduce a third author for text editing in the HART dataset. We designed experiments to explore the impact of different editor sequences on the final text.
>
> DETree is trained on RealBench and evaluated using the development set as the retrieval database. First, we investigated the influence of the first author on the final text, while keeping the second and third authors constant. The results show that when the third author is introduced, the model's ability to distinguish the first author is reduced. However, the model can still effectively differentiate between the two text types.
>
> |source1|source2|DETree(aucroc)|
> |-|-|-|
> |gpt-4o_humanize_gpt-3.5-turbo_polish_gemma-3|human_rephrase_gpt-3.5-turbo_polish_gemma-3|96.99|
> |gpt-4o_humanize_gpt-3.5-turbo_polish_deepseekv2|human_rephrase_gpt-3.5-turbo_polish_deepseekv2|96.21|
> |gpt-4o_humanize_gpt-3.5-turbo|human_rephrase_gpt-3.5-turbo|100|
> |gpt-4o_polish_gemma-3-12b|human_polish_gemma-3-12b|99.43|
> |gpt-4o_polish_deepseekv2|human_polish_deepseekv2|99.37|
> ||||
> |claude-3-5-sonnet_humanize_gemini-1.5-pro_polish_gemma-3|human_rephrase_gemini-1.5-pro_polish_gemma-3|84.17|
> |claude-3-5-sonnet_humanize_gemini-1.5-pro_polish_deepseekv2|human_rephrase_gemini-1.5-pro_polish_deepseekv2|84.05|
> |claude-3-5-sonnet_humanize_gemini-1.5-pro|human_rephrase_gemini-1.5-pro|98.14|
> |claude-3-5-sonnet_polish_gemma-3-12b|human_polish_gemma-3-12b|98.81|
> |claude-3-5-sonnet_polish_deepseekv2|human_polish_deepseekv2|98.98|
>
>
> We also examined the influence of the middle editor on the final text. The results show a significant decline in detection accuracy when the third author is introduced.
>
> |source1|source2|DETree(aucroc)|
> |-|-|-|
> |gpt-4o_humanize_gpt-3.5-turbo_polish_gemma-3|gpt-4o_humanize_claude-3-5-sonnet_polish_gemma-3-12b|76.50|
> |gpt-4o_humanize_gpt-3.5-turbo_polish_deepseekv2|gpt-4o_humanize_claude-3-5-sonnet_polish_deepseekv2|82.05|
> |gpt-4o_humanize_gpt-3.5-turbo|gpt-4o_humanize_claude-3-5-sonnet|98.78|
> ||||
> |human_rephrase_gpt-4o_polish_gemma-3|human_rephrase_gemini-1.5-pro_polish_gemma-3|68.71|
> |human_rephrase_gpt-4o_polish_deepseekv2|human_rephrase_gemini-1.5-pro_polish_deepseekv2|68.02|
> |human_rephrase_gpt-4o|human_rephrase_gemini-1.5-pro|99.49|
>
> Finally, the third editor is easily distinguishable.
>
> |source1|source2|DETree(aucroc)|
> |-|-|-|
> |qwen2.5-72b_humanize_claude-3-5-sonnet_polish_gemma-3|qwen2.5-72b_humanize_claude-3-5-sonnet_polish_deepseekv2|100|
> |llama-3.3-70b_humanize_gemini-1.5-pro_polish_gemma-3|llama-3.3-70b_humanize_gemini-1.5-pro_polish_deepseekv2|97.69|
> |human_rephrase_gpt-4o_polish_gemma-3|human_rephrase_gpt-4o_polish_deepseekv2|99.6|
>
> The initial creator of the text sets the general direction and ideas, so these features are difficult to mask through editing. However, the middle editor's features, which are based on rephrasing the initial text, are easily masked by the third editor, leading to a significant decrease in detection accuracy.
>
> This demonstrates that the DETree's encoder is capable of capturing fine-grained, generalizable features. Even when three authors are introduced into the test text (while the model was only exposed to two authors during training), and the third author's model, and prompt are unseen, the encoder can still extract key features to distinguish between different authors.
>
> > Weakness2&Q1: Theorem 3.1 is not stated as a theorem. The sentence "categories with closer LCA are expected to be more similar in the embedding space." is problematic because it is unclear over whay is the epxectation. The word "should" in the second line is confusing. Can you remove it? In Line 247 you refer to the "constraint defined in Theorem 3.1". What constraint? do you mean "condition"?
>
> We acknowledge that Theorem 3.1 was not clearly stated as a theorem, and we will revise it to a more formal mathematical expression. Specifically:
>
> If in the tree $\mathcal{T}$, for any leaf class X corresponding to node c, the following holds:
>
> $\mathbb{E}\bigl[\mathrm{sim}(X, Y)\bigr] > \mathbb{E}\bigl[\mathrm{sim}(X, Z)\bigr],  \quad \forall\, 0 \le i < j \le d_c,\; Y \in H_c^{(i)},\; Z \in H_c^{(j)}.$
>
> Then, we can equivalently deduce that for any leaf classes X, Y, Z, the following inequality holds:
>
> $\mathbb{E}\bigl[\mathrm{sim}(X, Y)\bigr] > \mathbb{E}\bigl[\mathrm{sim}(X, Z)\bigr], \quad \text{if } d_{\mathrm{LCA}(X, Y)} > d_{\mathrm{LCA}(X, Z)}.$
>
> Intuitively, we aim to optimize the second set of inequalities, but it is difficult to optimize directly. Therefore, we transform it into the equivalent first set of inequalities for optimization.
>
> We would appreciate your feedback on whether this form is appropriate, or if presenting it as a theorem may not be necessary.
>
> In line 247, the "constraint" refers to the inequality constraint defined by Equation 2 in the original paper. We will update the wording accordingly to avoid confusion and ensure clarity.
>
> > Weakness3: In Retrieval-Based Few-Shots adaptation, it is unclear what you mean by "superficial features" and why their "ability to generalize across domains" is limited. Of course, perforamcne bottlneck due to differences between train and test cases is the fundamental challenge in learning. It is unclear how the approach taken here is different than standard training? Also, do you allow few-shots adaptation in competing methods?
>
> We thank the reviewer for pointing this out. The term "superficial features" is indeed imprecise. We refer to features learned by supervised binary classification methods, which are optimized solely for binary discrimination and may capture spurious but separable features in the training data, leading to poor generalization in OOD settings. We performed dimensionality reduction on features extracted by MAGE (a supervised binary method) on the Beemo dataset (OOD) and observed significant distributional overlap, indicating a lack of effective separation. This is what we mean by the "limit ability to generalize across domains." We will include the corresponding visualization in the final version.
>
> In contrast, DETree captures intrinsic affinities across arbitrary categories, guiding the model to learn the fine-grained differences and relationships between categories. For example, even when texts are generated by different models from the same prompt, or when the same model generates texts through paraphrasing, polishing, or continuation, we aim for the model to understand the inherent relationships and distinctions between them. Thus, even with a small number of support samples, our method still achieves robust source matching and classification.
>
> In Table 2 (right), we report the performance of all models with few-shot capability under the same support set; in Table 2 (left), we report the zero-shot results of all models without such capability under the same test setting.
>
> > Weakness4: Related works: It is worth noting that the problem of deecting collaboration between authors existed before the AI era, and some methods were developed there [...]
>
> Indeed, author collaboration detection has been an active area of research prior to the AI era. We will incorporate discussions on text stylometry and author attribution in the related works section of the final version. Our approach, however, specifically focuses on detecting a wide range of human-AI collaborative text patterns in the context of the AI generation era. While we acknowledge the interpretability advantages of stylometry methods, our approach demonstrates superior generalization and robustness, offering a valuable complement to traditional methods.
>
> > Q2 Please explain if you allow few-shot adaptation when evaluating competing methods. This is crucial to understanding whether the improvement is the result of the proposed hierarchical contrastive methodology.
>
> In Table 1, we provide multiple sets of supervised experimental results, including training individually on each dataset and training with different priors on RealBench, which clearly demonstrates the performance improvements attributed to our hierarchical contrastive learning method.
>
> To ensure fairness, in Table 2, for models that support few-shot adaptation, we report their performance under the same support set. For models that do not support few-shot adaptation, we also provide results under the same testing environment without few-shot adaptation.

---

> > ### Author Response · Authors · 2025-08-06
> >
> > Dear Reviewer 1CCF,
> >
> > I hope this message finds you well. As the discussion period is nearing its end, I wanted to ensure we have addressed all your concerns satisfactorily. If there are any additional points or feedback you'd like us to consider, please let us know. Your insights are invaluable to us, and we remain eager to resolve any remaining issues to further improve our work.
> >
> > Thank you again for your time and effort in reviewing our paper.
> >
> > Best wishes,
> >
> > The Authors

---

> > > ### Comment · Reviewer_1CCF · 2025-08-06
> > >
> > > Regarding Theorem 3.1, unclear what is meant by "we can equivalently deduce that". Can you remove this phrase from the theorem while keeping it true?
> > >
> > > Regarding superficial features. Checking for distribution overlap after dimensionality reduction is not a good way to deduce that the features are useless, because there is a loss of information in this reduction.
> > >
> > > Regarding other works. The references I mentioned do not use stylistic features as a prediction baseline, but crossentropy loss under a certain language model. This seems to me very much related to the kind of detection used in this work.
> > >
> > > Regarding few-shot adaptation. It is unclear from your response what is going on in Table 1. In Table 2, it seems like only one method supports few-shot adaptation. Suppose that I suspect the reported superior performance of DETree is due to the unfair few-shot adaptation. Can you please refute this suspicion?

---

> > > > ### Author Response · Authors · 2025-08-07
> > > > **(Part 1/2) Response to Q1–Q3**
> > > >
> > > > > Q1: Regarding Theorem 3.1, unclear what is meant by "we can equivalently deduce that". Can you remove this phrase from the theorem while keeping it true?
> > > >
> > > > Thank you for pointing this out. We agree that the phrase "we can equivalently deduce that" is unnecessary. The revised version is as follows:
> > > >
> > > > If in the tree $\mathcal{T}$, for any leaf class X corresponding to node c, the following holds:
> > > >
> > > > $\mathbb{E}\bigl[\mathrm{sim}(X, Y)\bigr] > \mathbb{E}\bigl[\mathrm{sim}(X, Z)\bigr],  \quad \forall\, 0 \le i < j \le d_c,\; Y \in H_c^{(i)},\; Z \in H_c^{(j)}.$
> > > >
> > > > Then, for any leaf classes X, Y, Z, the following inequality holds:
> > > >
> > > > $\mathbb{E}\bigl[\mathrm{sim}(X, Y)\bigr] > \mathbb{E}\bigl[\mathrm{sim}(X, Z)\bigr], \quad \text{if } d_{\mathrm{LCA}(X, Y)} > d_{\mathrm{LCA}(X, Z)}.$
> > > >
> > > > > Q2: Regarding superficial features. Checking for distribution overlap after dimensionality reduction is not a good way to deduce that the features are useless, because there is a loss of information in this reduction.
> > > >
> > > > Although dimensionality reduction introduces some information loss, features extracted by DETree under the same reduction show clear separation, while those from supervised binary classification exhibit significant distribution overlap. Combined with classification metrics, the difference is evident. Therefore, we consider the features from supervised binary classification to be superficial and tightly tied to the training data, whereas DETree features are relatively more robust. We will include the corresponding visualization in the final version.
> > > >
> > > > > Q3: Regarding other works. The references I mentioned do not use stylistic features as a prediction baseline, but crossentropy loss under a certain language model. This seems to me very much related to the kind of detection used in this work.
> > > >
> > > > As you mentioned, early studies [1] extensively explored using language model-based metrics such as Perplexity, Entropy, and Rank for classification. However, recent work DetectGPT [2] showed that directly relying on these metrics yields suboptimal performance. DetectGPT hypothesizes that AI-generated texts lie at local extrema of the perplexity curvature, and small perturbations to such texts significantly increase their perplexity. Based on this, it proposes using the ratio of language model probabilities before and after perturbation as the classification signal, leading to substantial performance gains. FastDetectGPT[3] and Binoculars[4] further improve upon DetectGPT, achieving better results in both speed and accuracy.
> > > >
> > > > Therefore, we compare with Binoculars in Table 1, and with FastDetectGPT and Binoculars in Tables 2 and 3. These methods are also cited in the related work section (lines 74–75). Additionally, in HART (Table 3), we include experiments using Perplexity and Rank as classification signals, where perplexity is computed using GPT-Neo-2.7B, a model commonly used for such evaluation. The evaluation metric is TPR@5%FPR.
> > > >
> > > > ||Level-3|Level-2|Level-1|
> > > > |-|-|-|-|
> > > > |Log-Perplexity|33|11|6|
> > > > |Log-Rank|39|11|8|
> > > > |DETree|95.3|98.5|99.5|
> > > >
> > > > [1] Savoy J. Machine learning methods for stylometry[J]. Cham: Springer, 2020.
> > > >
> > > > [2] Mitchell E, Lee Y, Khazatsky A, et al. Detectgpt: Zero-shot machine-generated text detection using probability curvature[C]//International conference on machine learning. PMLR, 2023: 24950-24962.
> > > >
> > > > [3] Bao G, Zhao Y, Teng Z, et al. Fast-detectgpt: Efficient zero-shot detection of machine-generated text via conditional probability curvature[J]. arXiv preprint arXiv:2310.05130, 2023.
> > > >
> > > > [4] Hans A, Schwarzschild A, Cherepanova V, et al. Spotting llms with binoculars: Zero-shot detection of machine-generated text[J]. arXiv preprint arXiv:2401.12070, 2024.

---

> > > > ### Author Response · Authors · 2025-08-07
> > > > **(Part 2/2) Response to Q4**
> > > >
> > > > >Q4: Regarding few-shot adaptation. It is unclear from your response what is going on in Table 1. In Table 2, it seems like only one method supports few-shot adaptation. Suppose that I suspect the reported superior performance of DETree is due to the unfair few-shot adaptation. Can you please refute this suspicion?
> > > >
> > > > We think this is a misunderstanding of our experimental setup. As clearly stated in lines 241–243 of the paper, Table 1 presents supervised detection results, while Table 2 focuses on OOD generalization. Therefore, only in Table 2, we compare the few-shot capability of our method using OOD data in new domains. All methods in Table 1 are evaluated under standard testing settings.
> > > >
> > > > Although our method is retrieval-based, not all evaluations are under the few-shot setting. In most cases, we use the training set as the retrieval database. Only under the few-shot setting, we replace the database with a small number of samples from the same distribution as the test set.
> > > >
> > > > In Table 1, we compare with supervised learning methods, all trained on the training set and evaluated on the corresponding test set of each dataset. The "w/ per dataset" setting for DETree follows the same setup. The ablation "w/o TSCL" shows a notable performance drop after removing the Tree-Structured contrastive learning. Details are provided in the caption of Table 1.
> > > >
> > > > In Table 2, we report results under both zero-shot and few-shot settings.
> > > >
> > > > For zero-shot settings, MAGE OOD setting, DetectRL and Beemo are entirely unseen test sets for all methods, so for Fast-Detect, Binoculars, MAGE, RADAR, and DETree (database), these evaluations fall under the zero-shot setting. Since DETree is a retrieval-based method, it is trained on RealBench and tested using RealBench, MAGE, M4, and RAID as retrieval databases. MAGE, M4, and RAID are subsets of RealBench and thus do not violate the zero-shot setting. (Additionally, MAGE-unseen and MAGE-paraphrase are two OOD test scenarios specifically designed within MAGE, with both models and domains not present in the training data.)
> > > >
> > > > UAR and DETree (few-shot) are evaluated under the same few-shot setting.
> > > >
> > > > Zero-shot and few-shot results are reported separately. Therefore, our comparison setting is fair.
> > > >
> > > > Therefore, the suspicion that DETree’s superior performance comes from few-shot data is incorrect.

---

> > > > > ### Comment · Reviewer_1CCF · 2025-08-07
> > > > >
> > > > > Thank you for this explanation, which settles this matter.

---

### Official Review · Reviewer_UQj1 · 2025-07-03

**Clarity:** 2
**Significance:** 3
**Originality:** 3
**Rating:** 5
**Confidence:** 4

**Summary:**

DETree encodes texts with a supervised contrastive objective, constructs a Hierarchical Affinity Tree (HAT) from class centroids, and re-optimises representations with a tree-structured contrastive loss; at test time it performs k-NN retrieval over a K-means–compressed bank of only 10K prototypes, cutting memory and latency while preserving accuracy.
Across the HART hierarchy, RealBench and MAGE benchmarks, DETree attains up to 0.988 F1 / 99.5% TPR@5 % FPR, beating the strongest published detector by 30–40 pp on several sub-tasks. The model maintains high accuracy when trained on 10% of categories and gains ~30 pp in few-shot out-of-distribution transfer. Without any adversarial training DETree remains markedly more robust to diverse perturbations than an adversarially-trained RoBERTa baseline.

**Questions:**

* It would be nice to show how “human-like” the automatically generated edits are. For example: measuring the stylistic difference between AI-polished text and human-polished text (perhaps using a language model evaluator or some linguistic metrics).
* For the most part the paper is clearly written, but a few points could use clarification. The hierarchical prior assumptions are introduced in text and Figure 7, but the main paper only briefly references “prior1/2/3” in results without deep explanation. Readers must dig into Appendix B to understand them. Also, some implementation details, such as the exact temperature used in TSCL or how the *K*\-NN search is accelerated (if at all), are not obvious in the main text. The authors could consider adding them as footnotes.
* Lots of prior work on detection uses TPR@1% FPR as the metric, as keeping the number of false positives low is critical. Could the authors compute and report some results with this metric instead of TPR@5% FPR?

**Ethical Concerns:**

["NO or VERY MINOR ethics concerns only"]

**Final Justification:**

The rebuttal addressed my points raised, and I choose to maintain my original positive rating.

**Limitations:**

yes

**Quality:**

3

**Strengths And Weaknesses:**

Strengths

* The paper introduces a Hierarchical Affinity Tree (HAT) to model relationships among text generation processes, moving beyond flat binary or multi-class detectors. This is novel in that it does not treat "hybrid" text as just another class or rely on coarse metrics of AI involvement.
* The proposed RealBench dataset automatically curates \~16.4 million samples across 1,204 categories of human-AI collaborative text, which is far larger and more diverse than prior benchmarks. RealBench covers a wide spectrum of collaboration modes (human drafts refined by AI, AI outputs edited by humans, AI outputs refined by other AI models, etc.) and concrete transformation strategies including paraphrasing, extension, polishing, and translation.
* Experimental scope is broad, covers baselines from 12 detector families and 8 distinct benchmarks (including in- and out-of-distribution).

Weaknesses

* One concern is the manual intervention used in shaping the HAT. The authors acknowledge that purely agglomerative clustering can yield "unnatural" splits, so they employ an editable top-down reorganization using task-specific prior knowledge (Appendix B). In practice, they experimented with three hand-crafted priors (hybrid text closer to AI, closer to human, or separate) and chose the one yielding best performance. While this analysis is interesting, it does mean the final tree is partly guided by human assumptions rather than learned entirely from data.
* Two-stage training on a 16M-sample corpus plus k-NN retrieval at inference raises memory/latency concerns despite prototype compression.
* The paper focuses on texts generated through one-step human-AI collaboration, e.g. a human draft then one AI edit, or one AI generates then one human edit, etc. It mentions not covering scenarios with more than two "authors" due to data scarcity. This is a reasonable limitation, but it means very complex yet realistic writing processes (iterative back-and-forth edits or multiple AIs) are out of scope.

---

> ### Author Rebuttal · Authors · 2025-07-31
>
> *We sincerely thank the reviewer UQj1 for the thoughtful and detailed comments. Below, we provide point-by-point responses to each of the concerns.*
> ***
> > Weakness1: One concern is the manual intervention used in shaping the HAT. The authors acknowledge that purely agglomerative clustering can yield "unnatural" splits, so they employ an editable top-down reorganization using task-specific prior knowledge (Appendix B). In practice, they experimented with three hand-crafted priors (hybrid text closer to AI, closer to human, or separate) and chose the one yielding best performance. While this analysis is interesting, it does mean the final tree is partly guided by human assumptions rather than learned entirely from data.
>
> As mentioned, the use of manual intervention in constructing the HAT is guided by task-specific prior knowledge to make slight adjustments, enhancing the model's fit for the Human-AI Collaborative Texts detection task. However, it is important to note that the tree structure is primarily driven by the similarity matrix learned in the first stage of the model. We believe that one of the key strengths of DETree lies in its ability to balance data adaptiveness with task flexibility, allowing for moderate optimization that aligns with the task while preserving the underlying hierarchical relationships from the data.
>
> > Weakness2: Two-stage training on a 16M-sample corpus plus k-NN retrieval at inference raises memory/latency concerns despite prototype compression.
>
> During the training phase, we use a RoBERTa-large encoding model with approximately 0.5B parameters. Given the large dataset size, training can be completed in about 40 hours using 8×RTX4090 GPUs.
>
> In the inference phase, we utilize a smaller, non-autoregressive encoding model, which is faster compared to methods that calculate text probability distributions using large language models. When we compress the database to 10k samples for k-NN retrieval classification, the computational overhead is equivalent to attaching a 10k-sized classification head (~10M parameters) after the encoder.
>
> Binoculars and Fast_DetectGPT run inference on two RTX 4090 GPUs, while RADAR, MAGE, and DETree run inference on a single RTX 4090 GPU. We tested inference speed with 1k samples and report the average inference time per sample.
>
> ||Time Pre Smaple|
> |-|-|
> |Binoculars|136.5ms|
> |Fast_DetectGPT|132.9ms|
> |MAGE|47.6 ms|
> |RADAR|18.9ms|
> |DETree|26.9ms|
>
> > Weakness3: The paper focuses on texts generated through one-step human-AI collaboration, e.g. a human draft then one AI edit, or one AI generates then one human edit, etc. It mentions not covering scenarios with more than two "authors" due to data scarcity. This is a reasonable limitation, but it means very complex yet realistic writing processes (iterative back-and-forth edits or multiple AIs) are out of scope.
>
> Thank you for highlighting this limitation. Due to resource constraints, we have not previously explored scenarios involving more than two "authors." However, when sufficient data is available, our method can be directly extended to handle more complex writing processes, such as iterative back-and-forth edits or collaborations involving multiple AIs. To better align with realistic scenarios involving more than two authors, we have expanded the HART dataset to include three authors and conducted the following experiments:
>
> We used models not seen during training, such as gemma-3-12b and deepseekv2, along with new prompts to introduce a third author for text editing in the HART dataset. We designed experiments to explore the impact of different editor sequences on the final text.
>
> DETree is trained on RealBench and evaluated using the development set as the retrieval database. First, we investigated the influence of the first author on the final text, while keeping the second and third authors constant. The results show that when the third author is introduced, the model's ability to distinguish the first author is reduced. However, the model can still effectively differentiate between the two text types.
>
> |source1|source2|DETree(aucroc)|
> |-|-|-|
> |gpt-4o_humanize_gpt-3.5-turbo_polish_gemma-3|human_rephrase_gpt-3.5-turbo_polish_gemma-3|96.99|
> |gpt-4o_humanize_gpt-3.5-turbo_polish_deepseekv2|human_rephrase_gpt-3.5-turbo_polish_deepseekv2|96.21|
> |gpt-4o_humanize_gpt-3.5-turbo|human_rephrase_gpt-3.5-turbo|100|
> |gpt-4o_polish_gemma-3-12b|human_polish_gemma-3-12b|99.43|
> |gpt-4o_polish_deepseekv2|human_polish_deepseekv2|99.37|
> ||||
> |claude-3-5-sonnet_humanize_gemini-1.5-pro_polish_gemma-3|human_rephrase_gemini-1.5-pro_polish_gemma-3|84.17|
> |claude-3-5-sonnet_humanize_gemini-1.5-pro_polish_deepseekv2|human_rephrase_gemini-1.5-pro_polish_deepseekv2|84.05|
> |claude-3-5-sonnet_humanize_gemini-1.5-pro|human_rephrase_gemini-1.5-pro|98.14|
> |claude-3-5-sonnet_polish_gemma-3-12b|human_polish_gemma-3-12b|98.81|
> |claude-3-5-sonnet_polish_deepseekv2|human_polish_deepseekv2|98.98|
>
> We also examined the influence of the middle editor on the final text. The results show a significant decline in detection accuracy when the third author is introduced.
>
> |source1|source2|DETree(aucroc)|
> |-|-|-|
> |gpt-4o_humanize_gpt-3.5-turbo_polish_gemma-3|gpt-4o_humanize_claude-3-5-sonnet_polish_gemma-3-12b|76.50|
> |gpt-4o_humanize_gpt-3.5-turbo_polish_deepseekv2|gpt-4o_humanize_claude-3-5-sonnet_polish_deepseekv2|82.05|
> |gpt-4o_humanize_gpt-3.5-turbo|gpt-4o_humanize_claude-3-5-sonnet|98.78|
> ||||
> |human_rephrase_gpt-4o_polish_gemma-3|human_rephrase_gemini-1.5-pro_polish_gemma-3|68.71|
> |human_rephrase_gpt-4o_polish_deepseekv2|human_rephrase_gemini-1.5-pro_polish_deepseekv2|68.02|
> |human_rephrase_gpt-4o|human_rephrase_gemini-1.5-pro|99.49|
>
> Finally, the third editor is easily distinguishable.
>
> |source1|source2|DETree(aucroc)|
> |-|-|-|
> |qwen2.5-72b_humanize_claude-3-5-sonnet_polish_gemma-3|qwen2.5-72b_humanize_claude-3-5-sonnet_polish_deepseekv2|100|
> |llama-3.3-70b_humanize_gemini-1.5-pro_polish_gemma-3|llama-3.3-70b_humanize_gemini-1.5-pro_polish_deepseekv2|97.69|
> |human_rephrase_gpt-4o_polish_gemma-3|human_rephrase_gpt-4o_polish_deepseekv2|99.6|
>
> The initial creator of the text sets the general direction and ideas, so these features are difficult to mask through editing. However, the middle editor's features, which are based on rephrasing the initial text, are easily masked by the third editor, leading to a significant decrease in detection accuracy.
>
> This demonstrates that the DETree's encoder is capable of capturing fine-grained, generalizable features. Even when three authors are introduced into the test text (while the model was only exposed to two authors during training), and the third author's model, and prompt are unseen, the encoder can still extract key features to distinguish between different authors.
>
> > Q1: It would be nice to show how “human-like” the automatically generated edits are. For example: measuring the stylistic difference between AI-polished text and human-polished text (perhaps using a language model evaluator or some linguistic metrics).
>
> We supplement analysis by using language model evaluation metrics (Log-Perplexity, Log-Rank) on the HART dataset to measure the stylistic similarity between AI-polished and human-polished texts. Specifically, for the two types of texts, we use these evaluation metrics as the basis for classification. If the two types of texts cannot be separated effectively (i.e., low classification accuracy), it indicates that the distributions of the texts are highly consistent under the given metric.
>
> HART uses the commonly used model GPT-Neo-2.7B to provide the conditional probability of each token. Log-Perplexity is calculated as follows:
>
> $\text{Log-Perplexity}=-\frac{1}{N}\sum_{i=1}^{N}\log p(w_i\mid w_{<i})$
>
> Log-Rank measures the position of the correct token in the predicted ranking. For each token, we record its rank $r_i$ in the probability-sorted list, then take the logarithm and compute the average:
>
> $\text{Log‑Rank}=\frac{1}{t}\sum_{i=1}^{t}\log r_{\theta}\bigl(x_i\mid x_{<i}\bigr)$
>
> We report the classification metric TPR@5% FPR. The detailed description of the HART dataset can be found in lines 749-761 of the paper. The differences between AI-polished and human-polished texts using these metrics are visible in Level-2 of the HART dataset.
>
> ||Level-3|Level-2|Level-1|
> |-|-|-|-|
> |Log-Perplexity|33|11|6|
> |Log-Rank|39|11|8|
> |DETree|95.3|98.5|99.5|
>
> > Q2: For the most part the paper is clearly written, but a few points could use clarification. The hierarchical prior assumptions are introduced in text and Figure 7, but the main paper only briefly references “prior1/2/3” in results without deep explanation. Readers must dig into Appendix B to understand them. Also, some implementation details, such as the exact temperature used in TSCL or how the KNN search is accelerated (if at all), are not obvious in the main text. The authors could consider adding them as footnotes.
>
> Regarding the hierarchical prior assumptions, we will provide a more detailed explanation of these in the final version to ensure that readers can better understand our assumptions. Additionally, we will include further details on the temperature (0.05) used in TSCL and the Faiss-GPU acceleration of the K-NN search, providing clear explanations as footnotes.
>
> > Q3: Lots of prior work on detection uses TPR@1% FPR as the metric, as keeping the number of false positives low is critical. Could the authors compute and report some results with this metric instead of TPR@5% FPR?
>
> Since the HART paper uses TPR@5% FPR as the comparison metric, we adopted TPR@5% FPR for consistency in our evaluation. We have now included the performance of DETree with different priors on the HART dataset using TPR@1% FPR.
>
> ||Level-3||Level-2||Level-1||
> |-|-|-|-|-|-|-|
> ||TPR@5%FPR|TPR@1%FPR|TPR@5%FPR|TPR@1%FPR|TPR@5%FPR|TPR@1%FPR|
> |prior1|95.3|88.9|98.5|88.4|99.5|96.7|
> |prior2|88.7|76.0|93.6|58.7|98.9|85.8|
> |prior3|92.2|83.1|96.8|84.1|99.7|94.0|

---

> > ### Author Response · Authors · 2025-08-06
> >
> > Dear Reviewer UQj1,
> >
> > I hope this message finds you well. As the discussion period is nearing its end, I wanted to ensure we have addressed all your concerns satisfactorily. If there are any additional points or feedback you'd like us to consider, please let us know. Your insights are invaluable to us, and we remain eager to resolve any remaining issues to further improve our work.
> >
> > Thank you again for your time and effort in reviewing our paper.
> >
> > Best wishes,
> >
> > The Authors

---

> > ### Comment · Reviewer_UQj1 · 2025-08-07
> >
> > Thank you for addressing my comments in detail! The results presented in the rebuttal would be valuable additions to the paper.

---

### Decision · Program_Chairs · 2025-09-17

**Decision:**

Accept (poster)

**Comment:**

This paper tackles the timely and practically important problem of detecting hybrid human–AI authorship by learning a hierarchical provenance structure (HAT) and aligning embeddings with a tree‑structured contrastive loss (TSCL). While the building blocks (contrastive learning, clustering, k‑NN) are familiar, their combination for hierarchical provenance is apt and yields strong empirical gains across supervised, OOD, few‑shot, and hybrid evaluations. The release of RealBench (16.4M samples, 1204 categories) is a significant resource that will likely catalyze follow‑up work. Main concerns are (i) manual priors and subtree edits during HAT construction (risk of bias/selection), (ii) the synthetic nature of RealBench and confirmed cross‑split duplicates in one source dataset, and (iii) reliance on a retrieval database at inference that can blur OOD boundaries if not carefully set. The rebuttal addressed measurement and fairness questions. Reviewer consensus is favorable (three accepts, one low‑confidence borderline with a clear misreading).